# Autocomp: LLM-Driven Code Optimization for Tensor Accelerators

Charles Hong, Sahil Bhatia, Alvin Cheung, Yakun Sophia Shao

UC Berkeley

{*charleshong, sahilbhatia, akcheung, ysshao*}*@berkeley.edu*

*Abstract*—Hardware accelerators, especially those designed for tensor processing, have become ubiquitous in today's computing landscape. However, even with significant efforts in building compilers, programming these tensor accelerators remains challenging, leaving much of their potential underutilized. Recently, large language models (LLMs), trained on large amounts of code, have shown significant promise in code generation and optimization tasks, but generating low-resource languages like specialized tensor accelerator code still poses a significant challenge. We tackle this challenge with Autocomp, an approach that empowers accelerator programmers to leverage domain knowledge and hardware feedback to optimize code via an automated LLM-driven search. We accomplish this by: 1) formulating each optimization pass as a structured two-phase prompt, divided into planning and code generation phases, 2) inserting domain knowledge during planning via a concise and adaptable optimization menu, and 3) integrating correctness and performance metrics from hardware as feedback at each search iteration. Across three categories of representative workloads and two different accelerators, we demonstrate that Autocomp-optimized code runs 5.6× (GEMM) and 2.7× (convolution) faster than the vendor-provided library, and outperforms expert-level hand-tuned code by 1.4× (GEMM), 1.1× (convolution), and 1.3× (fine-grained linear algebra). Additionally, we demonstrate that optimization schedules generated from Autocomp can be reused across similar tensor operations, improving speedups by up to 24% under a fixed sample budget.

## I. Introduction

Hardware accelerators [29], [43] have become a critical driving force for the recent breakthroughs [18], [19], [32], [44], [59] in machine learning. They provide hundred-fold improvements in performance and energy efficiency in running deep neural networks (DNNs), and this has led to an increasing number of accelerators for tensor processing in recent years [7], [8], [13], [27], [29], [34], [42], [46]. However, extracting that performance requires writing high-performance accelerator code, which is time consuming and requires a deep understanding of the underlying hardware.

To address this challenge, various compilers and domain-specific languages (DSLs) have appeared. For deep learning applications, compilers such as XLA, TVM, and Triton generate high-performance code, but they only support a few hardware backends, particularly CPUs and GPUs [6], [45]. Unfortunately, adapting compilers and DSLs to new hardware platforms with vendor-specific instruction set architectures (ISAs) and implementation-specific dataflow patterns requires significant engineering effort. In fact, software alone comprises 40-50% of the development cost for new hardware [23], [54],

[58], even before considering the effort needed for end users to write and debug software for a newly developed chip. Prior work in DSLs like Halide and Exo [26], [50] targets accelerators by providing primitives that make it easier to express tensor computation, but the onus of optimizing code written in such DSLs still lies on the accelerator programmer.

Even once a compiler exists, generating performant code runs into the classical "scheduling" problem, i.e., deciding which optimizations to apply and in what order. For general-purpose backends (CPUs and GPUs), these passes have been iteratively developed and refined over many years through a combination of experts and auto-tuning frameworks. Recent work has gone further in exploring data-driven approaches such as supervised learning [66] and reinforcement learning [10], and even LLM training [9] to tackle the combinatorial explosion of pass sequences. While these data-driven approaches have shown promise, they depend on vast amounts of performance data to train, which is painfully scarce for domain-specific hardware accelerators.

In this paper, we present Autocomp, which solves the problems with prior approaches with an iterative LLM-guided search framework to optimize accelerator code. Unlike previous compilers, Autocomp can adapt to new hardware platforms and ISAs by simply changing prompts. Unlike existing tensor DSLs, Autocomp automatically generates optimized code without manual tuning. And unlike data-driven approaches targeting CPUs and GPUs, Autocomp requires no model training, instead leveraging LLM in-context reasoning and pretrained knowledge of common optimizations.

In each iteration, Autocomp first *plans* by choosing an optimization from a predefined menu, i.e., a list of common hardware accelerator optimizations like tiling and unrolling, then *applies* the optimization to generate optimized DSL code. The generated candidates are validated for correctness and benchmarked on the accelerator to collect performance metrics, providing feedback for the next iteration of search. By encoding DSL syntax, optimization rules, and performance feedback concisely in a prompt, Autocomp can guide the LLM to generate optimized code to run on the target accelerator.

In our evaluation, we apply Autocomp to two representative low-resource accelerators and generate code that runs 5.6× (GEMM) and 2.7× (convolution) faster than the vendor-provided library. Furthermore, it outperforms expert-level hand-tuned code by 1.4× (GEMM), 1.1× (convolution), and 1.3× (fine-grained linear algebra), surpassing the prior best

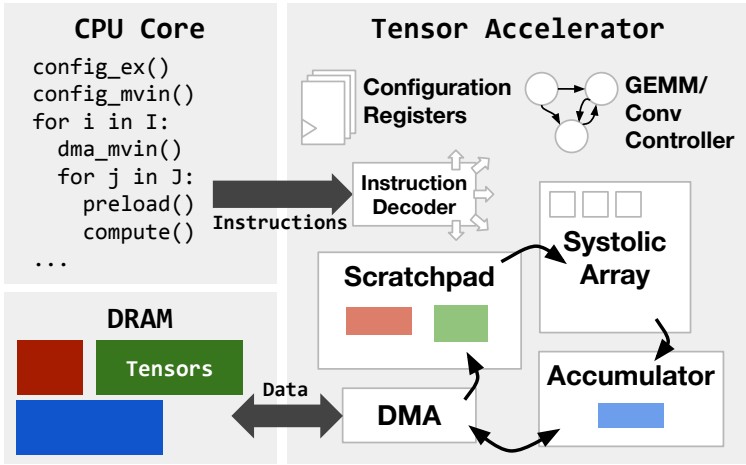





Fig. 1: Architecture and dataflow of a tensor accelerator system. Note that data movement is handled explicitly via accelerator direct memory access (DMA) instructions. GEMM/conv controllers refer to on-chip hardware implementations of matrix multiplications and convolutions, used as baselines in Sec. IV.

```
1   // CPU code
2   for (int i = 0; i < N; i++)
3     for (int j = 0; j < N; j++) {
4       C[i][j] = 0;
5       for (int k = 0; k < N; k++)
6         C[i][j] += A[i][k] * B[k][j];}
7
8   // Accelerator code
9   for (int ii = 0; ii < N; ii += T)
10  for (int jj = 0; jj < N; jj += T) {
11    zero_accumulator(acc_addr+...);
12    for (int kk = 0; kk < N; kk += T) {
13      dma_mvin(A[ii*T][kk*T], A_spad_addr);
14      dma_mvin(B[kk*T][jj*T], B_spad_addr);
15      for (int i = 0; i < T; i+=16)
16        for (int j = 0; j < T; j+=16) {
17          for (int k = 0; k < T; k+=16)
18            compute(A_spad_addr+..., B_spad_addr+...,
19                    acc_addr+...);}}}
20    dma_mvout(acc_addr, C[ii*T][jj*T]);}
```

Fig. 2: Comparison of general-purpose CPU code and tensor accelerator code for matrix multiplication.

known implementations with less human effort. Moreover, we show that Autocomp's schedules can be reused as guidance when scheduling similar tensor operations, alleviating the cost of scheduling new code and delivering up to 24% greater speedups under a fixed sample budget.

In summary, we make the following contributions in this paper:

1) We present Autocomp, the first LLM-driven code optimization approach for low-resource tensor accelerator code generation.
2) Autocomp incorporates domain knowledge, hardware feedback on correctness and performance, and novel strategies for response diversity to automatically generate performant code.
3) Our generated implementations significantly outperform hand-optimized code written by experts across a wide range of workloads and across different tensor accelerators.
4) We illustrate that schedules generated by Autocomp can be reused to optimize similar tensor operations, reducing search cost and demonstrating the *a posteriori* usefulness of Autocomp-generated schedules beyond pure performance.

## II. BACKGROUND

### A. Code Optimization for Tensor Accelerators

Programming tensor accelerators differs greatly from programming general-purpose CPUs. Tensor accelerators, depicted in Fig. 1, generally focus on the efficient execution of fixed-size (e.g., 16×16) matrix multiplication instructions, as shown in Fig. 2. Rather than trying to reduce the number or type of these instructions, which is often fixed, software optimization focuses on other aspects, such as:

- Minimizing data movement between main memory and smaller accelerator-local memories (in Fig. 1, the scratchpad and accumulator).
- Setting configuration state for computation and data movement.
- Scheduling or reordering operations to maximally overlap computation and data movement.

Code transformations that enable these optimizations range from low-level changes like arithmetic simplification or instruction selection, to higher-level changes like loop tiling, hoisting (Fig. 3), or software pipelining (Fig. 4). These high-level changes, while improving performance, require loop nests, pointers, and indices to be modified in multiple locations, making them challenging to implement, especially in a low-resource DSL.

Prior work has explored some of this optimization space. For example, performance models like Timeloop [48], MAE-STRO [33], and TeAAL [39] use high-level hardware architectural models and software abstractions to represent tensor accelerators and their workloads. Much recent work has sought to automatically explore this space, using methods such as machine learning [20], [22], [24], [52], linear programming [25], black-box optimization [30], [53], [65], and reinforcement learning [64]. While these abstractions capture some aspects of tensor accelerator code optimization, in particular the amount of data movement, they neglect other implementation-specific and instruction-level optimizations. In this work, LLM code generation allows us to directly rewrite accelerator code, expanding the search to include all potential axes of optimization.

### B. LLM-Based Code Optimization

LLMs have been used in various code-related tasks [2], [5], [41]. For code optimization, several works have utilized evolutionary approaches that take advantage of LLMs' abil-

```
1   // Unoptimized
2   for (int i = 0; i < 8; i++) {
3      for (int j = 0; j < 32; j++) {
4         for (int k = 0; k < 8; k++) {
5            config_mvin(128); // A's stride is 128
6            dma_mvin(A[i*16][k*16], spad_addr_1);
7            config_mvin(256); // B's stride is 256
8            dma_mvin(B[k*16][j*16], spad_addr_2);
9
10  // Optimized
11  config_mvin(128);
12  config_mvin_2(256);
13  for (int i = 0; i < 8; i++) {
14     for (int j = 0; j < 32; j++) {
15        for (int k = 0; k < 8; k++) {
16           dma_mvin(A[i*16][k*16], spad_addr_1);
17           dma_mvin_2(B[k*16][j*16], spad_addr_2);
```

Fig. 3: Example of hoisting accelerator configuration instructions, which can block execution. In this case the accelerator supports multiple direct memory access (DMA) load instructions, each with its own configuration state.

```
1   // Unoptimized
2   for (int k = 0; k < 8; k++) {
3      for (int i = 0; i < 32; i++) {
4         dma_mvin(A[i*16][k*64], spad_addr);
5         for (int k_i = 0; k_i < 4; k_i++) {
6            compute(spad_addr + k_i * 16, ...);
7
8   // Optimized
9   for (int k = 0; k < 8; k++) {
10     spad_addr = base_spad_addr;
11     dma_mvin(A[0][k*64], spad_addr);
12     for (int i = 0; i < 32; i++) {
13        dma_mvin(A[(i+1)*16][k*64], spad_addr + 64);
14        for (int k_i = 0; k_i < 4; k_i++) {
15           compute(spad_addr + k_i * 16, ...);
16        spad_addr += 64;
```

Fig. 4: Example of software pipelining in tensor accelerators. The A matrix tile is spread throughout accelerator memory rather than repeatedly loaded to the same location, allowing data loading to run ahead and overlap with computation.

ity to apply novel mutations to code [4], [16], [36]–[38], [51]. Others have used methods like in-context learning and retrieval-augmented generation (RAG) to transfer knowledge from a library of previously known optimizations to the current problem [1], [61]. Still others have collected large datasets of performance examples to fine-tune models [55], construct simplified program abstractions for faster optimization [62], or utilize compiler and runtime feedback to iteratively improve the generated code [11], [47], [49], [57].

Some prior work, for example Ouyang et al. [47] and Taneja et al. [57], targets system-level performance programming, specifically CUDA and SIMD intrinsics. However, we are not aware of any works that address LLM code optimization for specialized hardware (i.e., not CPUs or GPUs). Hong et al. [21] show that zero-shot code generation for such languages is highly unreliable. Nonetheless, Autocomp successfully optimizes accelerator code via a combination of novel techniques.

### III. THE **AUTOCOMP** APPROACH

#### A. Rationale

One naive way to generate optimized tensor accelerator code is to directly ask an LLM to rewrite the unoptimized code into its optimized counterpart. However, this approach fails for two reasons:

1) Tensor accelerator DSLs are low-resource languages (i.e., insufficiently represented in the LLM's training corpus), so the model produces neither semantically nor syntactically valid programs.
2) Without guidance, the model has little notion of what optimizations to apply, or in which order.

Prior work shows that decomposing tasks, including code generation tasks, into multiple steps can improve an LLM's ability to solve them [15], [21], [35], [60], [63]. Therefore, as shown in Fig. 5, we split our workflow into two phases: optimization plan generation and code implementation. We connect the two phases with an iterative beam search. Maintaining the best $B$ code candidates at each step helps us explore multiple optimization trajectories in parallel. We describe our

two phases and search strategy in the next section. Detailed prompts can be found in Appendix B.

In following sections, a *plan* refers to a natural language description of a single step of optimization and how code is transformed in that step, whereas a *schedule* refers to a sequence of plans that brings code from unoptimized to fully optimized.

#### B. Two-Phase Optimization

**Phase 1: Optimization selection and plan generation.** We prompt an LLM to select *one* optimization from a predefined menu of optimization options and to describe the concrete transformations required to apply it. Fig. 5 illustrates the prompt structure, which consists of the following parts:

1) **Accelerator ISA.** A list of instructions in the accelerator's ISA. We describe the semantics of each instruction in natural language, provide a specification for the accelerator's internal memory addresses, and briefly describe the accelerator's structure.
2) **Current Code.** In the initial iteration $t = 0$, the original unoptimized code. When $t > 0$, one of the $B$ candidates in our beam that has been selected for functional correctness and performance.
3) **Feedback.** The latency of **Current Code** in cycles, as well as its scratchpad and accumulator utilization in kilobytes. This helps the model choose the next optimization to apply. For example, low scratchpad utilization can lead to the model suggesting a larger tile transformation. Utilization is a common feedback metric across hardware platforms that reflects how effectively we are using the accelerator's hardware resources.
4) **Optimization Menu.** A list of high-level optimizations, for instance, *loop unrolling*, *reordering*, *fusion*, *tiling*, and *double buffering*. Note that only the names of the optimization are included; we rely on model to generate the implementation details for the selected optimization. The full list of optimizations is in Appendix B.

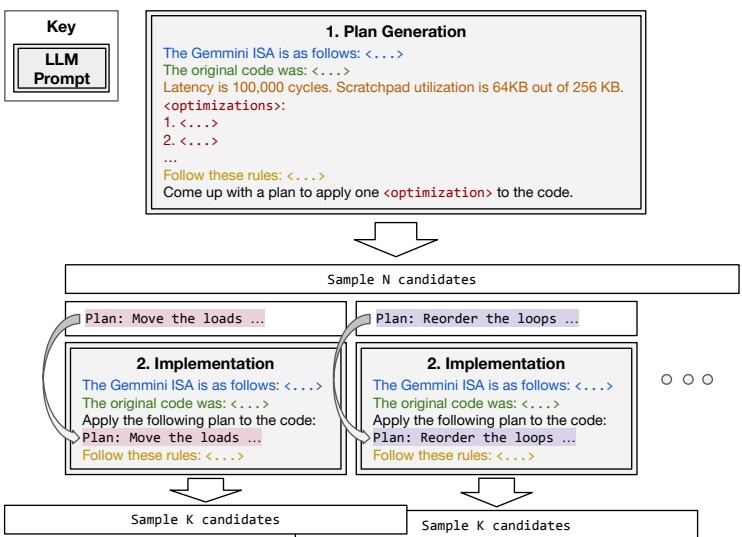

Fig. 5: Autocomp's two-phase optimization (Sec. III-B) and carried out at each iteration of beam search.

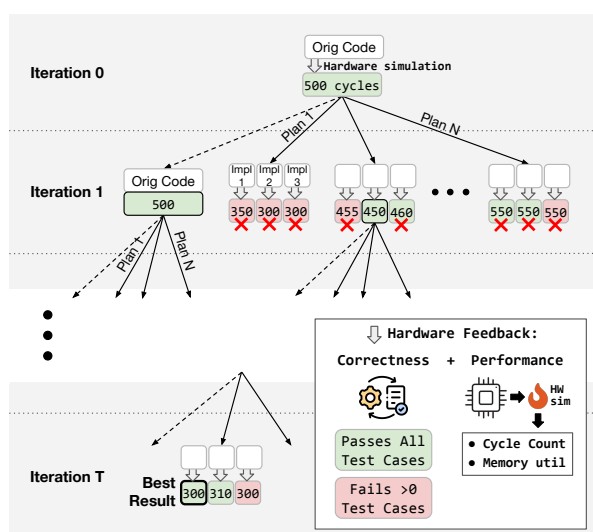

Fig. 6: Autocomp's beam search, described in Sec. III-C.

5) **Instruction.** A simple natural-language directive to "select exactly one optimization from the menu and output a corresponding transformation plan."
6) **Rules.** A set of constraints regarding the eventual code to be generated (see Appendix B).

At each planning iteration, we sample $N$ independent plans, seeding the search with multiple diverse optimization trajectories that can be evaluated in parallel.

**Phase 2: Optimized code generation.** Once we generate the candidate plans, for each plan we prompt the LLM to apply the transformations in the plan to generate a new, semantically equivalent code. In Fig. 5, we show the structure of our code generation prompt, which contains four parts:

1) **Accelerator ISA.** Same as in plan generation.
2) **Current Code.** Same as in plan generation.
3) **Generated Plan.** The specific optimization plan generated for this code in Phase 1.
4) **In-Context Learning (ICL) Example.** In cases where the optimization plan contains the string ``tiling'', we provide an example of code (from a different workload, and with variable names anonymized) before and after changing one tiling factor. Inserted as tiling is a key optimization that requires modifications across the program, making it challenging to implement.
5) **Instruction.** A simple natural-language instruction to "apply the above plan and output optimized accelerator code that is functionally equivalent to the current code."
6) **Rules.** Same as in plan generation.

We sample $K$ independent code candidates for each plan because generating low-resource accelerator code is challenging, and also because our task also requires applying nontrivial transformations to the code. Sampling multiple candidates helps improve the robustness of our search process.

### C. Beam Search

We integrate our two-phase optimization inside an iterative *beam search* of width $B$. Beam search allows us to efficiently explore several optimization trajectories in parallel. Since our code mostly consists of loop nests instead of sequential code, we find that merging candidates as in prior work [4], [16], [36]–[38], [51] is not suitable to tensor accelerator code. As illustrated in Fig. 6, candidates from the code generation step enter the beam only if they satisfy the criteria *correctness* and *performance*:

1) **Correctness.** After each code generation step, every candidate is compiled and run against our functional test suite. Each input variable is initialized with random values and after running, the candidate's output is compared to that of a reference implementation. We first filter candidates by compiling and testing 5 times via functional hardware simulation, and up to 20 times in cycle-accurate simulation, which functionally matches real hardware at a cycle-level granularity.
2) **Performance.** We measure the latency of functionally correct candidates via cycle-accurate simulation. A candidate is retained only if it improves upon the parent from which it was derived.

Of the functionally correct candidates, we keep the best (lowest latency) $B$ to seed the next iteration of beam search. Beam width is a hyperparameter, and empirically we found width $B = 6$ the sweet spot balancing search quality and time trade-off. We run this loop for a fixed budget of iterations.

### D. Increasing Plan and Code Diversity

We use the following two techniques to boost the diversity in plan (and in the case of LLM ensembling, code) generation and prevent the model from repeatedly selecting the same optimization:

- **Optimization Menu Dropout.** Inspired by methods for preventing neural networks from overfitting [56], we implement dropout in our optimization menu. Each time a plan candidate is generated, each menu option in the list of optimizations has a chance to be dropped.
- **LLM Ensembling.** Ensembling LLMs is known to improve diversity and quality of results [28]. To further increase the diversity of generated plans and code, in each case where multiple candidates are sampled, we divide these requests between different LLMs.

We ablate these techniques, along with other components of Autocomp, in Appendix A.

### E. Schedule Reuse

Running Autocomp's search starting with unoptimized code results in optimized code that outperforms all prior approaches, as we will discuss in Sec. IV. However, using this approach with every new software workload can be costly, as Autocomp involves multiple LLM invocations and hardware simulations. A natural question, then, is whether the schedules discovered for one workload can be used to accelerate the optimization of others. We draw inspiration from traditional optimized libraries like BLAS [3], where hand-tuned schedules are reused across GEMM shapes, and extend Autocomp with schedule reuse.

To do so, we first record the best known schedule for a particular tensor operation. Then, during planning for new GEMMs with the same aspect ratios or with two shared dimensions, rather than exploring the full menu, we prompt the LLMs to select specifically the menu options used in our recorded schedule, one at a time. As we are not exploring the full menu, we can use a smaller beam width and sample count, reducing both LLM calls and search time. After completing this lightweight search, we take the best-performing code so far and further refine it by invoking the full Autocomp search for a small number of iterations. This resembles the classic exploration-exploitation trade-off in optimization: by reusing a schedule we exploit a known high-quality schedule and avoid the initial exploration cost for a new workload.

## IV. EVALUATING AUTOCOMP WITHOUT SCHEDULE REUSE

We evaluate the effectiveness of Autocomp on three distinct types of workloads [1]: 1) matrix multiplication (GEMM) derived from ResNet-50, 2) convolution derived from ResNet-50, and 3) robotics code used for model-predictive control. For all experiments, we ensemble OpenAI's o3-mini and gpt-4o (via the OpenAI API Platform) for both phases, with temperature 1.0. Menu options are dropped out with 70% probability.

### A. Hardware Platform

We use Gemmini [17] to generate two different accelerators for evaluation. Gemmini is an accelerator generator that can generate systolic array- and vector-style tensor accelerators

[1]Benchmarks uploaded to
https://drive.google.com/drive/folders/1zK34x0tnzbQhcPjABxW-5uFbAKfoLGp6

with a wide range of data types and sizes. Gemmini is ideal for evaluating Autocomp as it: 1) generates accelerators that deliver performance comparable to commercial ones, 2) is open-source, enabling instantiation of different accelerator instances, user modifications to the software toolchain, and extraction of fine-grained performance feedback, and 3) supports fast and cycle-accurate hardware simulation via FireSim [31]. Like other accelerators, its low-resource nature eliminates data contamination and makes directly prompting LLMs challenging. We used AWS EC2 F1 instances and local AMD Alveo U250 FPGAs to run FireSim.

For the GEMM and convolution benchmarks in Secs. IV-C and IV-D, we use Gemmini to generate a $16\times16$ systolic array on 8-bit integer data type (accumulating in 32-bit), 256 KB scratchpad, and 64 KB accumulator, the same platform used by Ikarashi et al. [26]. For the fine-grained linear algebra benchmarks in Sec. IV-E, we generate a 32-bit floating point accelerator with a $4\times4$ systolic array, and the same scratchpad and accumulator sizes, as used by Dong et al. [14]. This information is provided in the **Instruction** portion of our planning prompt.

### B. Baselines

For the first two workload types, we compare Autocomp with four baselines, characterized in Table I:

1) **Gemmini's Software Library.** Gemmini ships with a software library that uses heuristics to tile and run GEMMs and convolutions on generated accelerators. As loop ordering is fixed and loop bounds, addresses, and indices must be computed at runtime, this implementation incurs significant software overhead and cannot fully utilize hardware resources.

2) **Exo Unoptimized.** Exo [26] is a DSL for tensor computation and scheduling. It comes with a basic compiler that emits functional GEMM or convolution code that can be executed on accelerators generated by Gemmini. However, without benchmark-specific optimization, performance is highly suboptimal, as hardware resources such as local memories tend to be underutilized.

3) **Exo Optimized.** In Ikarashi et al. [26], Exo's and Gemmini's developers spent significant effort manually writing and hand-tuning benchmark- and accelerator-specific schedules for each of the GEMM and convolution sizes in Figs. 7 and 8. This is the previous best known software implementation for these benchmarks.

4) **Hardware FSM.** Gemmini can generate accelerators with specialized hardware units for two coarse-grained operations: GEMM and convolution. These hardware units, implemented as finite state machines (FSMs), encode control sequences for each of these operations in hardware. If tuned correctly, the FSMs can exceed the theoretical maximum performance of any software-based implementation, as hardware is inherently parallel. However, this is accomplished at the cost of scheduling flexibility, as well as increased area, power, and hardware complexity. We use the hardware FSM as a reference for the highest achievable

| Approach | Performance | GEMM/conv size-agnostic | Supports workloads other than GEMM/conv | No power/ area cost |
|---|---|---|---|---|
| Generic library (e.g. Gemmini SW Lib, Exo Unopt) | Low | ✓ | ✗ | ✓ |
| Hand tuning (e.g. Exo Opt) | Medium/High | ✗ | ✓ | ✓ |
| Hardware FSM | High | ✓ | ✗ | ✗ |
| **Autocomp** | **High** | ✓ | ✓ | ✓ |

TABLE I: Qualitative comparison of Autocomp to baseline approaches.

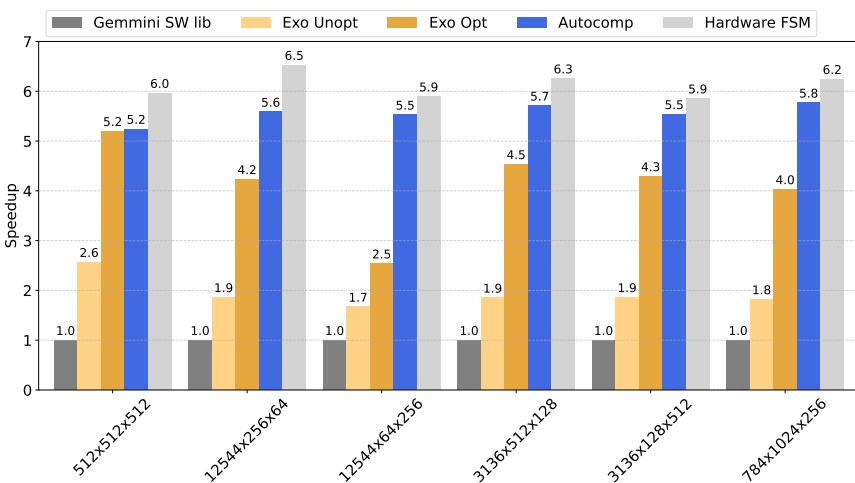

Fig. 7: Speedup for **GEMM** benchmarks.

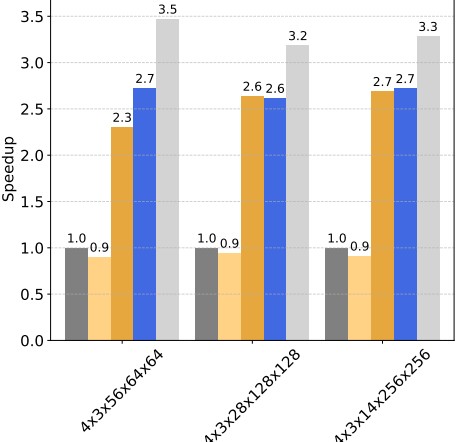

Fig. 8: Speedup for **convolution** benchmarks.

performance, but do not expect to exceed its performance for these GEMM/convolution benchmarks as its compute utilization exceeds 90% for all but one benchmark (82% on `12544x256x64`). Despite this, Autocomp-generated code approach hardware FSM performance for GEMM/convolution, and as seen in Sec. IV-E, even exceeds hardware FSM performance in end-to-end application performance thanks to Autocomp's greater scheduling flexibility.

For GEMM and convolution, we use Exo Unoptimized, which contains statically pre-computed loops, addresses, and indices, as Autocomp's starting point. This simplifies code generation and allows us to directly compare the effectiveness of Autocomp to hand-optimization (i.e., Exo Optimized code).

The third workload type, robotics control code, is a multi-kernel application containing sequences of element-wise operations and matrix-vector multiplications. As this is not directly supported by Exo, we compare to an unoptimized software implementation ported to accelerator code by Dong et al. [14], and an extensively hand-optimized hardware FSM-based implementation written by an expert.

### C. Matrix multiplication (GEMM)

We run Autocomp on a set of GEMM benchmarks selected by Ikarashi et al. [26] from ResNet-50 [19] for diversity in size and shape. We run search with beam size $B = 6$, $N = 6$ plans per element in the beam, $K = 2$ code candidates per plan, and $T = 15$ iterations. This takes around 5 hours to run.

Fig. 7 shows that Autocomp significantly outperforms even extensively hand-optimized code (Exo Opt) by a geomean of

$1.4\times$, Exo Unoptimized code (the starting point of Autocomp's search) by $2.9\times$, and Gemmini's software library by $5.6\times$. Autocomp is consistent: its generated code always achieves at least 85% of the hardware FSM's utilization (and 91% on average).

Autocomp especially outperforms prior implementations thanks to extensive exploration of software pipelining and double-buffering, which allows better overlapping of data movement and computation, for example by double-buffering both the scratchpad and accumulator. In many cases, Autocomp's exploration also leads to different tiling and loop ordering choices than hand-optimized code, reducing data movement. We qualitatively analyze Autocomp-generated GEMM code in detail in Appendix C.

### D. Convolution

We also optimize convolution benchmarks from ResNet-50 via the same process. Compared to the GEMM benchmarks, this code contains more loops and is more complex. In this case, we run beam search with beam size $B = 6$, $N = 12$ plans, and $K = 4$ code candidates, for $T = 10$ iterations, which takes about 7 hours.

Compared to GEMM, convolution provides less room for improvement over both the Gemmini software library and Ikarashi et al. [26]'s implementation, as even the hardware FSM only achieves a $3.3\times$ geomean speedup over the software library, compared to $6.1\times$ for GEMM. This is because on average the Gemmini software library achieves 28% of the theoretical maximum compute utilization, compared to 16% for

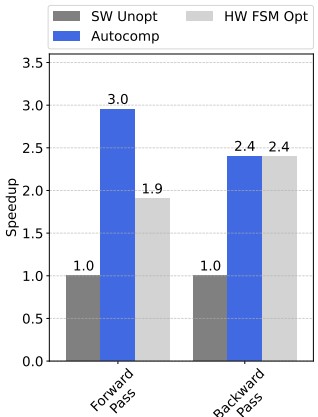

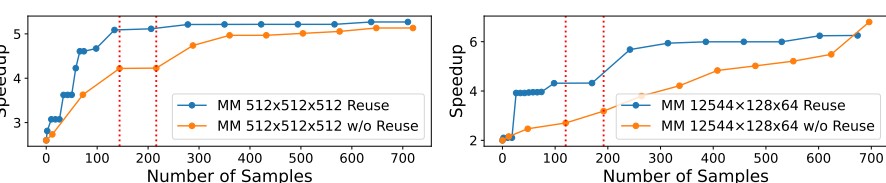

| Category | Base Benchmark | Reuse Targets |
|---|---|---|
| Square | 1024×1024×1024 | 512×512×512, 256×256×256 |
| Column-dominant | 12544×256x64 | 6272×256x64, 12544×128×64 |
| Row-dominant | 128×1024×1024 | 64×1024×1024, 32×1024×1024 |

TABLE II: GEMM benchmarks for schedule reuse experiments.

Fig. 9: Speedup for **fine-grained linear algebra** benchmarks.

Fig. 10: With the same sample budget, Autocomp with reuse (blue line) consistently delivers improved performance over Autocomp without reuse (orange line).

GEMM. As discussed by Genc et al. [17], at the tensor sizes in ResNet-50, convolutions have greater arithmetic intensity than GEMMs, making them less memory-bound and causing the Gemmini software library's suboptimal data orchestration to be less impactful.

Nonetheless, as shown in Fig. 8, Autocomp still exceeds the previous best known hand-optimized software ISA-based implementation (Exo Opt) by up to $1.2\times$ and by a geomean of $1.1\times$, via similar strategies as for GEMM. It also outperforms Exo Unoptimized code by $2.9\times$, and Gemmini's software library by $2.6\times$, and in all cases achieves at least 78% of the hardware FSM's utilization.

### E. Fine-Grained Linear Algebra

Finally, we optimize fine-grained linear algebra benchmarks from the TinyMPC model-predictive control library [40], specifically the forward and backward passes of the primal update step. These benchmarks contain sequences of floating-point matrix-vector multiplications, interleaved with element-wise addition and subtraction. The inclusion of CPU-accelerator dependencies, low reuse, and a high ratio of data movement to computation leads to low accelerator utilization and makes this code challenging to optimize.

We compare Autocomp-generated code against Dong et al. [14]'s unoptimized software-based implementation on a $4\times4$ FP32 accelerator. For this work, we additionally had an expert hand-tune a hardware FSM-based implementation. The unoptimized software-based implementation is used as the starting point for search, and we use the same search parameters as for convolution, except with $T = 15$ iterations, which takes about 12 hours. However, some of the optimization menu options are different from those used for GEMM/convolution (see Appendix B). As shown in Fig. 9, Autocomp outperforms even the expert-optimized hardware FSM implementation on the forward pass (by $1.6\times$), and across benchmarks speeds up unoptimized code by a geomean of $2.7\times$.

To outperform the hardware FSM implementation, Autocomp harnesses the flexibility of software-based implementation. It optimizes the code by hoisting data loads shared between kernels (reducing data movement beyond what is possible for the hardware FSM implementation), as well as utilizing fine-grained software pipelining and eliminating blocking operations where possible. This experiment highlights Autocomp's adaptability: we optimize a new benchmark, running on an accelerator with a new size and data type, with highly different performance characteristics from previous experiments, by updating the **Accelerator ISA** and changing a few lines in the **Optimization Menu** and **Instruction** sections of the prompt.

## V. CONCLUSION

In this work, we demonstrate that it is possible to construct an LLM-based flow to automatically optimize low-resource accelerator code at superhuman levels. Autocomp outperforms the previous best known software implementation by $1.34\times$ when generating matrix multiplication code, by $1.03\times$ for 2D convolution, and even outperforms a hardened matrix multiplication implementation by $1.51\times$ for a selection of robotics model-predictive control kernels. Autocomp's approach generates code which more accurately and consistently reflects the peak potential of a particular hardware design. This, combined with the potential to significantly reduce engineering effort, demonstrates Autocomp's potential to serve as a key component in the accelerator design process.

In future work, we would like to reduce the sample complexity, runtime, and monetary cost of our optimization flow. For example, we could do so by memorizing the sequence of optimizations applied to particular kernels, and applying a similar sequence of optimizations to similar kernels, by hand or using methods such as RAG.

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

| Experiment | 12544x64x256 GEMM Speedup | 4x3x14x256x256 Conv Speedup |
|---|---|---|
| Baseline (Exo Unopt) | 1.67× | 0.91× |
| No Accelerator ISA | 3.11× | 2.51× |
| No Optimization Menu | 2.34× | 0.97× |
| No Optimization Menu Dropout | 4.72× | 2.30× |
| No LLM Ensemble (o3-mini only) | 4.67× | 2.08× |
| No Hardware Perf Feedback | 4.91× | 2.61× |
| LLM Selection (DeepSeek-R1) | 4.89× | 2.25× |
| **Autocomp** | **5.53×** | **2.72×** |

TABLE III: Speedup relative to Gemmini's software library for each of the studies in this section. We include two representative benchmarks, one GEMM and one convolution, from our initial evaluation.

In this section, we ablate various features of Autocomp to investigate their effect on optimization performance. We focus on two specific benchmarks from Sec. IV—our `12544x64x256` GEMM and our `4x3x14x256x256` convolution—to isolate the effects of these ablations while limiting the cost of running this extensive exploration.

### A. Accelerator ISA

We find that for GEMM and convolution code, removing the ISA significantly deteriorates performance. Still, Autocomp is able to improve over the original code by a notable margin even without the ISA, given that all its other features are still enabled (see Table III). This is because we inherently provide an example of accelerator ISA code at each step via the current code, so the model is able to infer some properties of the accelerator ISA. In addition, many of the nested loop optimizations for the GEMM and convolution workloads are well-understood transformations that operate on the loops, addresses, and indexes in the code, which resemble general-purpose programming, rather than using accelerator-specific constructs such as the configuration instructions. However, full Autocomp performance is not matched as the proportion of functionally correct responses is lower, and instruction-level optimizations cannot easily be identified. For example, the first-compute handling optimization in Appendix C-A and the negative-scaled bias loading in Appendix C-E would not have been identified without knowledge of the ISA. Overall, we find that the accelerator ISA is an important part of Autocomp's prompts.

### B. Optimization Menu

We ablate the optimization menu by removing the menu in the planning prompt, and instead simply asking the model to select one optimization and generate a plan. From this experiment, we find that adding domain knowledge and optimization diversity via the optimization menu is essential to Autocomp. As shown in Table III, optimization performance completely deteriorates without the optimization menu. Qualitatively, without the optimization menu, we find that the models tend to repeat similar optimizations, with significantly less diversity and relevance in the generated optimization plans.

### C. Optimization Menu Dropout

"Dropout" for optimization menu options is a key contribution of this for increasing the diversity of generated optimization plans. Table III shows that menu dropout has a significant effect on performance. Qualitatively, we find that without dropout, models tend to be biased towards a limited set of menu options, a limitation which can be resolved via menu dropout.

### D. Hardware Performance Feedback

As discussed in Sec. III-B, during plan generation, we include the latency, scratchpad utilization, and accumulator utilization of the original code. Table III shows that this component is helpful, but in some cases its effects may be limited. This is because the options listed in the optimization menu already capture some of the metrics measured in our performance feedback, for example the menu options which suggest using larger tile sizes. Hardware feedback such as scratchpad and accumulator utilization only serves to augment elaboration of these menu options by providing exact measurements.

### E. LLM Ensembling

Splitting requests between LLMs in an ensemble also encourages diversity of generated plans and code. Qualitatively, we find that the responses, especially during the planning phase, generated by different LLMs differ substantially. Our experiments in Table III show that using individual models, such o3-mini or DeepSeek-R1, on their own results in significantly lower performance.

## F. LLM Selection

In Sec. IV, we use an ensemble of gpt-4o and o3-mini for our search. To demonstrate that Autocomp does not depend on a particular family of models, we run Autocomp with DeepSeek-R1 [12] on the same benchmarks used for other ablation experiments above. We use DeepSeek-R1 for both the planning and code generation phases and keep the search parameters identical to those used for matrix multiplication and convolution in Sec. IV. As shown in Table III, Autocomp with DeepSeek-R1 is able to optimize both GEMM and convolution, achieving substantial speed-ups over the unoptimized code. These experiments do not use LLM ensembling. Similarly to the LLM ensembling ablation study above, gains are slightly smaller than when all techniques are applied. Nonetheless, this demonstrates that Autocomp is efficient and flexible across different LLMs.

## APPENDIX B
## PROMPTS

```
#define config_ex(dataflow, act, A_stride, A_transpose, B_transpose)
// configure the state of the accelerator
// dataflow is WEIGHT_STATIONARY or OUTPUT_STATIONARY
// act is the activation function, options are NO_ACTIVATION, RELU, LAYERNORM, IGELU, SOFTMAX
// A_stride is the stride with which rows of A in the scratchpad are loaded into the systolic array, during computes. If this stride is 1, then we feed consecutive rows in
        the scratchpad, starting from the starting address of A, into the systolic array as the A matrix. If the stride is 2, then we feed every other row into the systolic
        array instead.
// A_transpose is a boolean value that represents whether the matrix A is transposed
// B_transpose is a boolean value that represents whether the matrix B is transposed

#define config_ld(dram_stride, scale_factor, spad_block_stride, id)
// configure mvin instructions
// dram_stride = stride in bytes, with which to load from DRAM
// scale_factor = factor to multiply loaded values
// spad_block_stride = when more than DIM columns are loaded, the distance in rows between each block of DIM columns
// id = id of mvin instruction; id = 0 for mvin, 1 for mvin2, 2 for mvin3

#define mvin(dram_addr, spad_acc_addr, cols, rows)
// mvin from DRAM to scratchpad or accumulator
// mvin, configured by config_ld(..., 0)
// rows must be less than or equal to DIM. if more than DIM rows, multiple mvin instructions are needed
// cols must be less than or equal to 4 * DIM.
// if dram_addr = 0, then zeroes are moved into scratchpad/accumulator, max size DIM x DIM

#define mvin2(dram_addr, spad_acc_addr, cols, rows)
// behavior identical to mvin, but configured by config_ld(..., 1)

#define mvin3(dram_addr, spad_acc_addr, cols, rows)
// behavior identical to mvin, but configured by config_ld(..., 2)

// A = input matrix, B = weight matrix, C = output matrix
// assume a weight-stationary dataflow
// preload, compute_preloaded, and compute_accumulated are used to compute DIM x DIM matrix multiplications.
// if no bias, C = A * B is computed; if there is a bias, C = A * B + bias is computed

#define preload(B_spad_addr, C_acc_addr, B_cols, B_rows, C_cols, C_rows)
// preload weights, B, onto DIM by DIM systolic array
// B must be preloaded before compute
// B must have been moved in to the scratchpad first
// B_cols must be less than or equal to DIM, B_rows must be less than or equal to DIM, C_cols must be less than or equal to DIM, C_rows must be less than or equal to DIM
// must run to change the output address to C_acc_addr
// if B_spad_addr unchaged from previous preload instruction, can set B_spad_addr = 0xffffffff; must be specified otherwise

#define compute_preloaded(A_spad_addr, bias_spad_addr, A_cols, A_rows, bias_cols, bias_rows)
// compute on DIM by DIM systolic array, with optional added bias (can be used for matrix addition)
// A must have been moved in to the scratchpad first
// first compute after preload to systolic array
// either overwrites or accumulates C depending on bit 30 of C_acc_addr
// A_cols must be less than or equal to DIM, A_rows must be less than or equal to DIM, bias_cols must be less than or equal to DIM, bias_rows must be less than or equal to
        DIM
// bias_spad_addr = 0xffffffff if no bias
// if there is a bias, bias_cols and bias_rows are probably equal to C_cols and C_rows from preload instruction

#define compute_accumulated(A_spad_addr, bias_spad_addr, A_cols, A_rows, bias_cols, bias_rows)
// compute on DIM by DIM systolic array
// A must have been moved in to the scratchpad first
// for weight stationary, use when B_spad_addr has not changed
// either overwrites or accumulates C depending on bit 30 of C_acc_addr
// A_cols must be less than or equal to DIM, A_rows must be less than or equal to DIM, bias_cols must be less than or equal to DIM, bias_rows must be less than or equal to
        DIM
// bias_spad_addr = 0xffffffff if no bias
// if there is a bias, bias_cols and bias_rows are probably equal to B_cols and B_rows from preload instruction

#define config_st(cols)
// configure mvout instruction
// cols = number of columns of matrix in DRAM

#define mvout(dram_addr, spad_acc_addr, cols, rows)
// mvout from scratchpad or accumulator to DRAM
// cols must be less than or equal to DIM
// rows must be less than or equal to DIM

#define fence() asm volatile("fence")
// fence
```

Fig. 11: Accelerator ISA specification for Gemmini accelerators, referenced in Sec. III-B.

```
'''
Gemmini's private memory is "row-addressed", where each row is DIM elements wide, where DIM is the number of PEs across the width of the systolic array. These elements
    will be of type inputType in the scratchpad, and of type accType in the accumulator.

Every private Gemmini memory address is 32 bits long. The three most signficant bits are reserved, and have special meanings:

    Bit 31 (the MSB) is 0 if we are addressing the scratchpad, and 1 if we are addressing the accumulator.
    Bit 30 is ignored if we are addressing the scratchpad, or if we are reading from the accumulator. If, instead, we are writing to the accumulator, then bit 30 is 0 if
        we want to overwrite the data at that address, and 1 if we want to accumulate on top of the data already at that address.
    Bit 29 is ignored if we are addressing the scratchpad, or if we are writing to the accumulator. If, instead, we are reading from the accumulator, then bit 29 is 0 if
        we want to read scaled-down inputType data from the accumulator, and 1 if we want to read accType data from the accumulator.
        If bit 29 is 1 for an accumulator read address, then we do not apply activation functions or scaling to the output of the accumulator.
'''

'''
Gemmini is a decoupled access/execute architecture, which means that "memory-access" and "execute" instructions happen concurrently, in different regions of the hardware.
It has an ExecuteController (for preload and compute instructions), LoadController (mvin), and StoreController (mvout).
Gemmini includes an ROB which is meant to detect hazards between instructions in different controllers.
Each controller also handles its own dependencies and hazards internally.
'''
```

Fig. 12: Accelerator ISA specification from Sec. III-B, continued.

```
<optimizations>:
1. modify loop tiling
2. loop reordering
3. split loops
4. fuse loops
5. simplify arithmetic and propagate constants to simplify expressions
6. reorder computations or blocks of computations
7. loop unrolling
8. double buffering
9. move more data to the scratchpad in a more outer loop to increase data reuse
10. spread data throughout the scratchpad rather than loading to the same location repeatedly
11. load data to the scratchpad across outer loop iterations and use if statements to prevent redundant loads on loops
     inner to those
12. hoist redundant operations out of loops
13. substitute operations with equivalent operations that are faster
14. pipeline operations to better overlap computation and data movement
15. minimize data movement
16. minimize loop overhead
17. other methods not listed here.
```

Fig. 13: The list of optimizations menu options available (with some probability of dropout) during the planning phase, as described in Sec. III-B. This menu was used to optimize both matrix multiplication and convolution code.

```
<optimizations>:
1. remove unnecessary code
2. simplify arithmetic and propagate constants to simplify expressions
3. merge instructions
4. merge high-level operations
5. reorder operations or blocks of operations
6. move cpu-based computation to the accelerator
7. add or subtract a matrix using the bias
8. hoist redundant operations out of loops
9. substitute operations with equivalent operations that are faster
10. pipeline operations to better overlap computation and data movement
11. eliminate data dependencies and fence operations
12. minimize data movement
13. minimize loop overhead
14. other methods not listed here
```

Fig. 14: The optimization menu for TinyMPC code optimization.

```
Here is an example of increasing scratchpad tile size for the Y dimension of a 512x512 (X x Z) matrix A and 512x512 (Z x Y) matrix B multiplication. Original code:
    uint32_t b_offset = 16 * 16 * 4 * 8 * sizeof(int8_t);
    for (int_fast32_t y = 0; y < 8; y++) {
        uint32_t b_base_y = 64 * y;
        // Load B matrix slice
        for (int_fast32_t zo = 0; zo < 8; zo++) {
            uint32_t b_zo_offset = 4 * 16 * zo; // Number of columns per zo iteration
            for (int_fast32_t z = 0; z < 4; z++) {
                uint32_t b_index = ((zo * 4 + z) * ((16 * 4) * 16)) / 16; // Divide number of elements by 16 since scratchpad is row-indexed
                mvin3(&B[b_zo_offset + 16 * z][b_base_y], b_offset + b_index, 16 * 4, 16);
        }}
        for (int_fast32_t x = 0; x < 32; x++) {
            uint32_t res = 1 << 31;
            uint32_t a_base_x = 16 * x;
            // Load A matrix slice
            for (int_fast32_t zo = 0; zo < 8; zo++) {
                uint32_t a_index = (zo * (16 * 4) * 16) / 16;
                mvin2(&A[a_base_x][64 * zo], a_index, 16 * 4, 16);
            }
            // Computation
            for (int_fast32_t zo = 0; zo < 8; zo++) {
                uint32_t a_index = (zo * (16 * 4) * 16) / 16;
                for (int_fast32_t z = 0; z < 4; z++) {
                    uint32_t preload_flag = (zo == 0 && z == 0) ? 0 : 0x40000000;
                    for (int_fast32_t y_in_o = 0; y_in_o < 4; y_in_o++) {
                        uint32_t preload_index = ((zo * 4 + z) * ((16 * 4) * 16) + y_in_o * (16 * 16)) / 16; // Find correct scratchpad index to load B from
                        preload(b_offset + preload_index, res + (y_in_o * (16 * 16)) / 16 | preload_flag, 16, 16, 16, 16);
                        compute_preloaded(a_index + (z * (16 * 16)) / 16, ~((uint32_t)0), 16, 16, 16, 16);
                }}}
            // Store C matrix slice
            for (int_fast32_t y_in_o = 0; y_in_o < 4; y_in_o++) {
                mvout(&C[a_base_x][b_base_y + 16 * y_in_o], res + (y_in_o * (16 * 16)) / 16, 16, 16); // Divide number of elements by 16 since accumulator is row-indexed
        }}}
Retiled code
    uint32_t b_offset = 16 * 16 * 4 * 8 * sizeof(int8_t);
    for (int_fast32_t y = 0; y < 2; y++) { // Reduce number of y dimension outer loop iterations
        uint32_t b_base_y = 256 * y;
        // Load larger B matrix slice
        // Tiling reduces redundant loads of B matrix, reducing data movement and increasing data reuse
        for (int_fast32_t zo = 0; zo < 8; zo++) {
            uint32_t b_zo_offset = 4 * 16 * zo; // Number of columns per zo iteration
            for (int_fast32_t z = 0; z < 4; z++) {
                for (int_fast32_t y_in = 0; y_in < 4; y_in++) {
                    uint32_t b_index = (((zo * 4 + z) * 4 + y_in) * ((16 * 4) * 16)) / 16; // Divide number of elements by 16 since scratchpad is row-indexed
                    mvin3(&B[b_zo_offset + 16 * z][b_base_y + 64 * y_in], b_offset + b_index, 16 * 4, 16);
        }}}
        for (int_fast32_t x = 0; x < 32; x++) {
            uint32_t res = 1 << 31;
            uint32_t a_base_x = 16 * x;
            // Load A matrix slice
            // Tiling reduces redundant loads of A matrix, reducing data movement and increasing data reuse
            for (int_fast32_t zo = 0; zo < 8; zo++) {
                uint32_t a_index = (zo * (16 * 4) * 16) / 16;
                mvin2(&A[a_base_x][64 * zo], a_index, 16 * 4, 16);
            }
            // Computation
            for (int_fast32_t zo = 0; zo < 8; zo++) {
                uint32_t a_index = (zo * (16 * 4) * 16) / 16;
                for (int_fast32_t z = 0; z < 4; z++) {
                    uint32_t preload_flag = (zo == 0 && z == 0) ? 0 : 0x40000000;
                    for (int_fast32_t y_in_o = 0; y_in_o < 16; y_in_o++) { // Increase number of Y dimension inner loop iterations to increase tile size
                        uint32_t preload_index = (((zo * 4 + z) * 4) * ((16 * 4) * 16) + y_in_o * (16 * 16)) / 16; // Find correct scratchpad index to load B from
                        preload(b_offset + preload_index, res + (y_in_o * (16 * 16)) / 16 | preload_flag, 16, 16, 16, 16);
                        compute_preloaded(a_index + (z * (16 * 16)) / 16, ~((uint32_t)0), 16, 16, 16, 16);
                }}}
            // Store C matrix slice
            for (int_fast32_t y_in_o = 0; y_in_o < 16; y_in_o++) { // Move out a larger tile in the Y dimension
                mvout(&C[a_base_x][b_base_y + 16 * y_in_o], res + (y_in_o * 16 * 16) / 16, 16, 16); // Divide number of elements by 16 since accumulator is row-indexed
        }}}
```

Fig. 15: In-context learning example of tiling, provided during the code generation phase in Sec. III-B. Inserted in the prompt only when the string `"tiling"` is detected in the plan generated in Phase 1.

```
Rules:
1. The rewritten program should be semantically equivalent to the original program
2. Limit the scope of the plan to the selected optimization
3. All code must be inside the test() function
4. Do not use C preprocessing directives (#ifdef, #define, etc.)
5. If modifying loops, modify other related loop bounds and adjust address and index calculations to ensure the code is
     still correct
6. If increasing loaded tile size, ensure that data is spread throughout the scratchpad across all relevant dimensions
7. If loading across new dimensions, add the loop indices of those dimensions to scratchpad address calculations
8. If increasing loaded tile size, update preload and compute instructions to match the new data layout
9. If increasing loaded tile size, update base scratchpad addresses to fit new tile size
```

Fig. 16: The list of rules provided during both the planning and code implementation phases, as described in Sec. III-B.

# APPENDIX C
## CODE EXAMPLES

In this section, we discuss in greater depth what optimizations Autocomp applies in our evaluations and how Autocomp is able to achieve significantly better performance than hand-optimized code.

### A. *12544x64x256 GEMM*

Fig. 17 contains the unoptimized Exo-generated code, used as the starting point for search. Fig. 18 contains the code generated by Exo after hand-optimization by Ikarashi et al. [26]. Figs. 19 to 21 contain the result of Autocomp optimization on Exo Unoptimized code. While the code is substantially transformed from the original code, some aspects remain the same. For example, in this case the configuration instructions and loop ordering remain largely the same.

Of course, many optimizations have been applied to the code. We briefly summarize the optimization menu options selected and plans generated during the optimization process for this code. We also include the speedup after each respective optimization.

1) 1.67×: initial speedup of Exo Unoptimized code over Gemmini's software library before any optimization.
2) 1.93× after "hoist redundant operations out of loops". This plan hoists constants like `tile_dim = 16` and loop-invariant expressions like `ko * 64` and `k * 16` out of inner loops. These precomputed values are reused inside Gemmini ops in each iterations, so we should reduce the number of times they must be calculated.
3) 1.95× after "double buffering". This plan defines two buffer regions for matrices A and B in the scratchpad. A `buffer_toggle` flag is introduced to alternate between these buffers each iteration. All `mvin`, `preload`, and `compute` instructions are updated to use the active buffer based on the toggle. Data loading for the next iteration is scheduled earlier to overlap with current computation. Address calculations are adjusted to include buffer offsets accordingly.
4) 2.15× after "pipeline operations to better overlap computation and data movement". Moves `mvin2` (A tile load) to immediately after `compute_preloaded` in the ko loop to overlap A prefetch with current compute. Moves `mvin3` (B tile load) earlier in the k loop, before the next compute, to overlap B prefetch with current compute.
5) 3.13× after "load data to the scratchpad across outer loop iterations and use if statements to prevent redundant loads on loops inner to those". Adds a one-time load of the entire B matrix into a new scratchpad region (specifically, `new_B_base` = 8192) before the main `i` loop. Replaces repeated `mvin3` calls in inner loops with offset calculations into this preloaded B region.
6) 3.54× after "move more data to the scratchpad in a more outer loop to increase data reuse". Before the reduction (`k`) loop, load the entire 16x256 A-tile into scratchpad using four mvin2 calls to a new `A_tile_base` region. Remove the double-buffering (`a_toggle`) for A since the full tile is now resident. Replace nested `ko` and `k` loops with a single loop over 16 segments, computing each 16x16 A sub-tile address from `A_tile_base`. Adjust B tile `preload` and `compute_preloaded` calls accordingly, using a simpler tile index.
7) 4.87× after "double buffering". Restores double buffering by reserving two scratchpad regions: `A_tile_base0` and `A_tile_base1`, each holding one full 16x256 A-tile. In the outer `i` loop, alternate buffers using `i % 2` to select `current_buffer` and `next_buffer`. In each iteration (except last), while computing with `current_buffer`, issue `mvin2` to load the next A-tile into `next_buffer`. In the inner `compute` loops, use `current_buffer` for A-tile addresses. After `compute` and `mvout`, the next iteration uses the preloaded data.
8) 5.21× after "double buffering". Double-buffers accumulator by allocating two accumulator regions: `acc_base0` and `acc_base1`. In each `i` iteration, compute into `cur_acc_base` and `mvout` from `prev_acc_base` (except on first iteration), and swap `cur_acc_base` and `prev_acc_base` at the end of the loop.
9) 5.23× after "loop unrolling". Unrolls the innermost loop (`j_in_o`) by a factor of 4.
10) 5.53× after "fuse loops". "Fuses" loops by eliminating the loop over `j_in_o` where we `mvin` 0s to the accumulator. Instead, use the ability of `preload` to overwrite the values in the accumulator rather than accumulating, when beginning a new partial sum.

From this example, we observe that a diverse set of optimizations is selected, and that speedups are distributed throughout the optimization process rather than concentrated in just one or two steps, showing the importance of a well-designed iterative search process. From here, we summarize the differences between Autocomp-generated code and the previous best code (Exo Opt):

- **Tiling.** The Exo Opt code loads 128×256 tiles of A, whereas the Autocomp-generated code loads 32×256 tiles (divided into two 16×256 tiles) of A. While this means there is less reuse for the Autocomp-generated code, there is also less overhead needed for compute instructions to wait for each A tile to be loaded to the scratchpad. In combination with the rest of the optimizations applied by Autocomp, this leads to improved performance.
- **Double-buffering.** In the Autocomp-generated code, we see that both the scratchpad and accumulator are explicitly double-buffered. In the schedule, we can see that double buffering is applied 3 times. Initially (in step 3), both the A and B matrices (where matrix multiplication is represented as A×B=C), are double buffered in the scratchpad. However,

after steps 5 and 6, B and A (respectively) are no longer double-buffered as larger tiles are loaded before beginning computation. The accumulator is double-buffered in step 8, resulting in the code below. The Exo Opt code relies on the accelerator's out-of-order execution to handle executing mvin and mvout instructions without dependencies, ahead of the order in which they are issued.

- **Software pipelining.** The Autocomp-generated code explicitly issues A mvin instructions before they are needed for computation, whereas as above the Exo Opt code relies on hardware to handle overlapping of data movement and compute. Also, the Autocomp-generated code explictly issues all B mvin instructions at the beginning of the program, whereas the Exo Opt code interleaves these instructions with computation (but still loads the entire B matrix to the scratchpad, once overall). This does not have a significant impact on performance, but LLM-generated code is naturally biased towards such an implementation due to its simplicity.
- **First-compute handling.** The Autocomp-generated code utilizes the ability of compute instructions to overwrite the accumulator, whereas Exo Opt code explicitly issues mvin instructions to zero out the accumulator before beginning computation on a tile.
- **Arithmetic simplification.** Arithmetic on constants is fully simplified and handled inside shared variables wherever possible in the Autocomp-generated code, reducing the overhead of non-accelerator instructions.

Overall, we find that compared to the Exo Opt code, Autocomp-generated code applies more techniques to minimize the amount of CPU overhead during execution. The smaller tiles it uses, in combination with its more explicit application of double-buffering and software pipelining, results in highly tuned, fine-grained overlapping of data movement and computation and a very high level of performance.

*B. `12544x64x256` GEMM Unoptimized Code Example*

```
void test(int8_t A[12544][256], int8_t B[256][64], int8_t C[12544][64]) {
  config_st((64));
  config_ex(WEIGHT_STATIONARY, NO_ACTIVATION, 1, false, false);
  config_ld((64), 1.0f, 16, 2);
  config_ld((256), 1.0f, 16, 1);
  config_ld(0, 1.0f, 0, 0);

  for (int_fast32_t i = 0; i < 784; i++) {
    for (int_fast32_t j = 0; j < 1; j++) {
      uint32_t res = 1 << 31;
      for (int_fast32_t j_in_o = 0; j_in_o < 4; j_in_o++) {
        mvin( 0, res + ((j_in_o) * (256))/16,(16 + 0), (16 + 0) );
      }
      uint32_t a = 0;
      uint32_t b = 16 * 16 * 4 * 4 / 16;
      for (int_fast32_t ko = 0; ko < 4; ko++) {
        mvin2( &A[(16 * i)][64 * ko], a + ((ko) * (1024))/16, 16*(4 + 0), (16 + 0) );
        for (int_fast32_t k = 0; k < 4; k++) {
          mvin3( &B[(64 * ko + 16 * k)][64 * j], b + ((ko) * (4096) + (k) * (1024))/16, 16*(4 + 0), (16 + 0) );
        }
        for (int_fast32_t k = 0; k < 4; k++) {
          for (int_fast32_t j_in_o = 0; j_in_o < 4; j_in_o++) {
            preload(b + ((ko) * (4096) + (k) * (1024) + (j_in_o) * (256))/16, res + ((j_in_o) * (256))/16 | 0x40000000, (16 + 0), (16 + 0), (16 + 0), (16 + 0));
            compute_preloaded(a + ((ko) * (1024) + (k) * (256))/16, ~((uint32_t)0), (16 + 0), (16 + 0), 16, 16);
          }
        }
      }
      for (int_fast32_t j_in_o = 0; j_in_o < 4; j_in_o++) {
        mvout( &C[(16 * i)][16 * j_in_o + 64 * j], res + ((j_in_o) * (256))/16, (16 + 0), (16 + 0) );
      }
    }
  }
  fence();
}
```

Fig. 17: Example of Exo-generated unoptimized matrix multiplication code, from the experiments in Sec. IV-C. Achieves 28% utilization.

## C. `12544x64x256` GEMM Exo Optimized Code Example

```c
void test(int8_t A[12544][256], int8_t B[256][64], int8_t C[12544][64]) {
  config_st((64));
  config_ex(WEIGHT_STATIONARY, NO_ACTIVATION, 1, false, false);
  config_ld((64), 1.0f, 16, 2);
  config_ld((256), 1.0f, 16, 1);
  config_ld(0, 1.0f, 0, 0);

  uint32_t res = 1 << 31;
  uint32_t a = 0;
  uint32_t b = 16 * 16 * 4 * 4 * 8 * sizeof(int8_t) / 16;
  for (int_fast32_t io = 0; io < 98; io++) {
    for (int_fast32_t i = 0; i < 8; i++) {
      mvin( 0, res + ((i) * (1024))/16, (16), (16) );
      mvin( 0, res + ((i) * (1024) + 256)/16, (16), (16) );
      mvin( 0, res + ((i) * (1024) + (2) * (256))/16, (16), (16) );
      mvin( 0, res + ((i) * (1024) + (3) * (256))/16, (16), (16) );
      for (int_fast32_t ko = 0; ko < 4; ko++) {
        mvin2( &A[(16 * i + 128 * io)][64 * ko], a + ((i) * (4096) + (ko) * (1024))/16, 16*(4), (16) );
        if (io == 0) {
          if (i == 0) {
            mvin3( &B[(64 * ko)][0], b + ((ko) * (4096))/16, 16*(4), (16) );
          }
        }
        if (io == 0) {
          if (i == 0) {
            mvin3( &B[(16 + 64 * ko)][0], b + ((ko) * (4096) + 1024)/16, 16*(4), (16) );
          }
        }
        if (io == 0) {
          if (i == 0) {
            mvin3( &B[(32 + 64 * ko)][0], b + ((ko) * (4096) + (2) * (1024))/16, 16*(4), (16) );
          }
        }
        if (io == 0) {
          if (i == 0) {
            mvin3( &B[(48 + 64 * ko)][0], b + ((ko) * (4096) + (3) * (1024))/16, 16*(4), (16) );
          }
        }
        preload(b + ((ko) * (4096))/16, res + ((i) * (1024))/16 | 0x40000000, (16), (16), (16), (16));
        compute_preloaded(a + ((i) * (4096) + (ko) * (1024))/16, ~((uint32_t)0), (16), (16), 16, 16);
        preload(b + ((ko) * (4096) + 256)/16, res + ((i) * (1024) + 256)/16 | 0x40000000, (16), (16), (16), (16));
        compute_preloaded(a + ((i) * (4096) + (ko) * (1024))/16, ~((uint32_t)0), (16), (16), 16, 16);
        preload(b + ((ko) * (4096) + (2) * (256))/16, res + ((i) * (1024) + (2) * (256))/16 | 0x40000000, (16), (16), (16), (16));
        compute_preloaded(a + ((i) * (4096) + (ko) * (1024))/16, ~((uint32_t)0), (16), (16), 16, 16);
        preload(b + ((ko) * (4096) + (3) * (256))/16, res + ((i) * (1024) + (3) * (256))/16 | 0x40000000, (16), (16), (16), (16));
        compute_preloaded(a + ((i) * (4096) + (ko) * (1024))/16, ~((uint32_t)0), (16), (16), 16, 16);
        preload(b + ((ko) * (4096) + 1024)/16, res + ((i) * (1024))/16 | 0x40000000, (16), (16), (16), (16));
        compute_preloaded(a + ((i) * (4096) + (ko) * (1024) + 256)/16, ~((uint32_t)0), (16), (16), 16, 16);
        preload(b + ((ko) * (4096) + 1024 + 256)/16, res + ((i) * (1024) + 256)/16 | 0x40000000, (16), (16), (16), (16));
        compute_preloaded(a + ((i) * (4096) + (ko) * (1024) + 256)/16, ~((uint32_t)0), (16), (16), 16, 16);
        preload(b + ((ko) * (4096) + 1024 + (2) * (256))/16, res + ((i) * (1024) + (2) * (256))/16 | 0x40000000, (16), (16), (16), (16));
        compute_preloaded(a + ((i) * (4096) + (ko) * (1024) + 256)/16, ~((uint32_t)0), (16), (16), 16, 16);
        preload(b + ((ko) * (4096) + 1024 + (3) * (256))/16, res + ((i) * (1024) + (3) * (256))/16 | 0x40000000, (16), (16), (16), (16));
        compute_preloaded(a + ((i) * (4096) + (ko) * (1024) + 256)/16, ~((uint32_t)0), (16), (16), 16, 16);
        ...
        // Unrolling continues
      }
      mvout( &C[(16 * i + 128 * io)][0], res + ((i) * (1024))/16, (16), (16) );
      mvout( &C[(16 * i + 128 * io)][16], res + ((i) * (1024) + 256)/16, (16), (16) );
      mvout( &C[(16 * i + 128 * io)][32], res + ((i) * (1024) + (2) * (256))/16, (16), (16) );
      mvout( &C[(16 * i + 128 * io)][48], res + ((i) * (1024) + (3) * (256))/16, (16), (16) );
    }
  }
  fence();
}
```

Fig. 18: Example of hand-optimized matrix multiplication code from Ikarashi et al. [26], used as a baseline in Sec. IV-C. Achieves 43% utilization. Note that unrolled `preload`/`compute_preloaded` instructions have been truncated due to length.

## D. `12544x64x256` *GEMM Autocomp-Optimized Code Example*

```
void test(int8_t A[12544][256], int8_t B[256][64], int8_t C[12544][64]) {
  //------------------------------------------------------------------------
  // Tile and matrix constants
  //------------------------------------------------------------------------

  const uint32_t tile_dim   = 16;   // tile dimension
  const uint32_t tile_offset = tile_dim; // each row in a tile is tile_dim elements

  // For double buffering, we use two accumulator base addresses separated by 4 tile rows.
  const uint32_t acc_base0 = 1U << 31;
  const uint32_t acc_base1 = (1U << 31) + (4 * tile_dim);

  const uint32_t new_B_base   = 8192;
  const uint32_t A_tile_base0 = 2048;
  const uint32_t A_tile_base1 = 6144;

  //------------------------------------------------------------------------
  // Gemmini configuration
  //------------------------------------------------------------------------

  config_st(64);
  config_ex(WEIGHT_STATIONARY, NO_ACTIVATION, 1, false, false);
  config_ld(64, 1.0f, tile_dim, 2);
  config_ld(256, 1.0f, tile_dim, 1);
  config_ld(0, 1.0f, 0, 0);

  //------------------------------------------------------------------------
  // Load the entire weight matrix B into the scratchpad once.
  //------------------------------------------------------------------------

  for (int br = 0; br < 256; br += tile_dim) {
    mvin3(&B[br][0],
         new_B_base + (br / tile_dim) * (tile_dim * 4),
         tile_dim * 4,
         tile_dim);
  }

  //------------------------------------------------------------------------
  // Begin double-buffered accumulator computation.
  //
  // cur_acc_base: where the current tile is computed.
  // prev_acc_base: holds the result of the previously computed tile.
  //------------------------------------------------------------------------

  uint32_t cur_acc_base  = acc_base0;
  uint32_t prev_acc_base = 0; // Will be set after the first iteration.

  // Outer loop over tiles (784 tiles along A's first dimension)
  for (int i = 0; i < 784; i++) {

    // Determine which scratchpad region holds A for this tile.
    // Alternating between two buffers allows overlapping the load of the next tile.
    uint32_t current_A_buffer = (i % 2 == 0) ? A_tile_base0 : A_tile_base1;
    uint32_t next_A_buffer    = (i % 2 == 0) ? A_tile_base1 : A_tile_base0;

    //----------------------------------------------------------------
    // For the first iteration load the A tile into scratchpad.
    // For subsequent iterations, write the previous tile from the accumulator.
    //----------------------------------------------------------------
    if (i == 0) {
      for (int b = 0; b < 4; b++) {
        mvin2(&A[tile_dim * i][64 * b],
             current_A_buffer + b * (tile_dim * 4),
             tile_dim * 4,
             tile_dim);
      }
    } else {
      for (int j_in_o = 0; j_in_o < 4; j_in_o++) {
        uint32_t j_off = j_in_o * tile_dim;
        mvout(&C[tile_dim * (i - 1)][tile_dim * j_in_o],
             prev_acc_base + j_off,
             tile_dim,
             tile_dim);
      }
    }
```

Fig. 19: Example of an optimized version of the same GEMM from Fig. 17, generated using Autocomp. Achieves 93% compute utilization. Continued in Figs. 20 and 21.

```c
//-----------------------------------------------------------------------
// Instead of explicitly zeroing-out the accumulator tile via extended mvin,
// we fuse the accumulator zeroing into the compute stream.
//
// For each accumulator sub-tile (indexed by j_in_o), the very first compute
// call is issued in overwrite mode (i.e. the accumulator address is used as-is)
// and then all subsequent compute calls for that sub-tile are issued in accumulate mode.
//
// We use an array (first_compute) to track whether a given sub-tile has been updated.
//-----------------------------------------------------------------------
int first_compute[4] = { 1, 1, 1, 1 };

//-----------------------------------------------------------------------
// Compute the current tile.
// The complete computation is divided into 16 segments.
// Loop unrolling by a factor of 4 is applied to the segments loop.
//-----------------------------------------------------------------------
for (int seg = 0; seg < 16; seg += 4) {
  for (int j_in_o = 0; j_in_o < 4; j_in_o++) {
    uint32_t j_off = j_in_o * tile_dim;
    uint32_t acc_addr; // will hold the computed accumulator address for preload

    //--- Unrolled segment "seg" -------------------------------------------
    {
      uint32_t block_id  = seg / 4;
      uint32_t sub_offset = (seg % 4) * tile_dim;
      uint32_t A_seg_addr = current_A_buffer + block_id * (tile_dim * 4) + sub_offset;
      // If this is the very first compute for this sub-tile, use overwrite mode
      if (first_compute[j_in_o]) {
        acc_addr = cur_acc_base + j_off;
        first_compute[j_in_o] = 0;
      } else {
        acc_addr = (cur_acc_base + j_off) | 0x40000000;
      }
      preload(new_B_base + seg * (tile_dim * 4) + j_off,
              acc_addr,
              tile_dim, tile_dim,
              tile_dim, tile_dim);
      compute_preloaded(A_seg_addr,
                        ~(uint32_t)0,
                        tile_dim, tile_dim,
                        tile_dim, tile_dim);
    }

    //--- Unrolled segment "seg+1" -------------------------------------------
    {
      uint32_t block_id  = (seg + 1) / 4;
      uint32_t sub_offset = ((seg + 1) % 4) * tile_dim;
      uint32_t A_seg_addr = current_A_buffer + block_id * (tile_dim * 4) + sub_offset;
      preload(new_B_base + (seg + 1) * (tile_dim * 4) + j_off,
              (cur_acc_base + j_off) | 0x40000000,
              tile_dim, tile_dim,
              tile_dim, tile_dim);
      compute_preloaded(A_seg_addr,
                        ~(uint32_t)0,
                        tile_dim, tile_dim,
                        tile_dim, tile_dim);
    }

    //--- Unrolled segment "seg+2" -------------------------------------------
    {
      uint32_t block_id  = (seg + 2) / 4;
      uint32_t sub_offset = ((seg + 2) % 4) * tile_dim;
      uint32_t A_seg_addr = current_A_buffer + block_id * (tile_dim * 4) + sub_offset;
      preload(new_B_base + (seg + 2) * (tile_dim * 4) + j_off,
              (cur_acc_base + j_off) | 0x40000000,
              tile_dim, tile_dim,
              tile_dim, tile_dim);
      compute_preloaded(A_seg_addr,
                        ~(uint32_t)0,
                        tile_dim, tile_dim,
                        tile_dim, tile_dim);
    }
```

Fig. 20: Example from Fig. 19, continued.

```
      //--- Unrolled segment "seg+3" -----------------------------------------------
      {
        uint32_t block_id   = (seg + 3) / 4;
        uint32_t sub_offset = ((seg + 3) % 4) * tile_dim;
        uint32_t A_seg_addr = current_A_buffer + block_id * (tile_dim * 4) + sub_offset;
        preload(new_B_base + (seg + 3) * (tile_dim * 4) + j_off,
                (cur_acc_base + j_off) | 0x40000000,
                tile_dim, tile_dim,
                tile_dim, tile_dim);
        compute_preloaded(A_seg_addr,
                          ˜(uint32_t)0,
                          tile_dim, tile_dim,
                          tile_dim, tile_dim);
      }
    } // end inner loop over j_in_o

    //---------------------------------------------------------------------------
    // For seg==0 (i.e. the first unrolled iteration), launch prefetching of the next A tile.
    // This overlaps memory-access with computation.
    //---------------------------------------------------------------------------
    if (seg == 0 && i < 783) {
      for (int b = 0; b < 4; b++) {
        mvin2(&A[tile_dim * (i + 1)][64 * b],
              next_A_buffer + b * (tile_dim * 4),
              tile_dim * 4,
              tile_dim);
      }
    }
  } // end segments loop

  //---------------------------------------------------------------------------
  // Swap accumulator buffers.
  // The tile computed in this iteration (in cur_acc_base) becomes the previous tile,
  // so it must be written back in the next iteration.
  //---------------------------------------------------------------------------
  prev_acc_base = cur_acc_base;
  cur_acc_base = (cur_acc_base == acc_base0) ? acc_base1 : acc_base0;
} // end outer tile loop

//---------------------------------------------------------------------------
// Write back the final computed tile (tile index 783) from the accumulator.
//---------------------------------------------------------------------------
for (int j_in_o = 0; j_in_o < 4; j_in_o++) {
  uint32_t j_off = j_in_o * tile_dim;
  mvout(&C[tile_dim * (784 - 1)][tile_dim * j_in_o],
        prev_acc_base + j_off,
        tile_dim,
        tile_dim);
}

fence();
}
```

Fig. 21: Example from Figs. 19 and 20, continued.

*E. TinyMPC Primal Update Forward Pass (Fine-Grained Linear Algebra)*

The forward pass of the TinyMPC primal update step computes the following operations:

$$u[i] = -K_{inf} * x[i] - d[i]$$

$$x[i+1] = (A_{dyn} * x[i]) + (B_{dyn} * u[i])$$

Where $A_{dyn}$ is a `12x12` matrix, $B_{dyn}$ is a `12x4` matrix, $K_{inf}$ is a `4x12` matrix, $x$ is an `NHORIZONx12` matrix (where individual columns are accessed here via indexing), $d$ is an `NHORIZONx4` matrix, and $u$ is an `NHORIZONx4` matrix. $A_{dyn}$, $B_{dyn}$, $K_{inf}$, $d$ and the 0th column of $x$ are inputs, and $u$ is the output. The 1st to (NHORIZON-1)th column of $x$ are intermediate values computed over the course of the benchmark.

This process is repeated until a time horizon, defined as `NHORIZON` in our code and set to 5 for our evaluations. Note that $x$ is allocated as an `(NHORIZON+1)x12` matrix in our code since the unoptimized code accesses up to the `(NHORIZON)`th column.

Autocomp generates the code in Fig. 24, optimized over several steps from the starting code in Fig. 17. The following optimizations are applied, with the following speedups after each optimization:

1) $1\times$: in this case we treat the unoptimized software as the baseline for speedup, so by definition its speedup is $1\times$.
2) $1.07\times$ after "hoist redundant operations out of loops". Hoists the `mvin` calls for the constant matrices `Kinf`, `Adyn`, and `Bdyn` above the `NHORIZON` loop and executes them once rather than in every iteration. Any associated `config_ex` and `config_ld` calls are also moved outside the loop if needed. The compute calls use the same scratchpad addresses and may set `B_spad_addr = 0xffffffff` to indicate that the weights are already loaded.
3) $1.13\times$ after "loop reordering". This plan claims to move the configuration and `mvin` instructions for `Kinf`, `Adyn`, and `Bdyn` before the `NHORIZON` loop, but this has already been handled in the previous step. In practice, only some configuration instructions that were unnecessarily left behind by the previous step are hoisted, hence the limited improvement.
4) $2.00\times$ after "move CPU-based computation to the accelerator". Replaces CPU-based element-wise negation and addition with equivalent Gemmini compute instructions. Specifically, when `x_i` and `d_i` are loaded, a scaling factor of -1 is used to negate it in the scratchpad. The product of `Kinf` and `x_i` is kept in the accumulator and the negated `d_i` is multiplied by a dummy 0 vector in order accumulate it on top of this product, enabling data to be kept resident in the accumulator rather than moving it back and forth between the CPU and accelerator memory.
5) $2.12\times$ after "hoisting redundant operations out of loops". Identifies a few `config_ld` instructions that are still redundant and removes them from the loop.
6) $2.66\times$ after "loop unrolling". Changes the outer loop to increment by 2 instead of 1 and duplicate the loop body so each iteration computes two time steps: first u[i] and x[i+1], then u[i+1] and x[i+2]. Also merges `fence` calls at the end of the unrolled loop body if possible. The implementation very aggressively removes `fence` instructions, which is actually correct as reuse of the same accelerator memory instructions by subsequent `mvin` and `mvout` instructions means that dependencies can be handled internally to the accelerator, rather than via synchronization of both the CPU and accelerator.
7) $2.95\times$ after "loop fusion". Undoes the loop unrolling from the previous step, but keeps the reduced `fence` instructions. Additionally, this eliminates the unnecessary calculation of `x[NHORIZON]` during execution, which saves cycles. This optimization makes sense since it is likely the reduced number of `fence` instructions that improved performance in step 6, rather than the loop unrolling (which usually provides limited benefit, if any).

We further discuss the similarities and differences between the hand-optimized hardware FSM-based implementation in Fig. 23 and the Autocomp-optimized code:

- **Data orchestration.** Due to the coarse-grained nature of the hardware FSM operations and the fact that data movement between the accelerator and main memory is handled within hardware, we are not able to hoist shared data loads of the `Kinf`, `Adyn`, and `Bdyn` matrices out of the loop. This is the main advantage the Autocomp-generated software ISA-based implementation has over the hardware FSM-based implementation.
- **Operator fusion.** Both implementations are able to handle all computation on the accelerator, but notably the hardware FSM-based implementation fuses the addition of `d[i]` into the bias when multiplying `Kinf` and `x[i]` (then the whole result is negated while storing to main memory), whereas the addition of `d[i]` is handled as a separate compute instruction in the Autocomp-generated code. So, the hardware FSM-based implementation actually has an advantage in this regard and the Autocomp-generated code has further room for improvement. Both implementations make use of negative scaling of loaded data (via `config_ld` instructions) in order to handle subtraction.
- **Fence instructions.** The Autocomp-generated code is able to remove CPU-based fence instructions and instead handle dependencies within the accelerator, whereas the hardware FSM-based code is forced to place a fence after each matrix multiplication, as accumulated results must be moved back to main memory and loaded to the scratchpad for the next operation.

- **Configuration overhead.** While they do not have as much overhead as fence instructions, configuration instructions can also be blocking. The hand-optimized code hoists configuration instructions out of the loop where possible, but since different matrix sizes must be loaded inside the loop, configuration instructions cannot be completely eliminated, giving Autocomp's code the advantage in this aspect.
- **Dead code elimination.** Both implementations eliminate the extra computation of `x[NHORIZON]` that is present in the unoptimized code.

Overall, we find that Autocomp identifies all major optimization opportunities available in the code, with the exception of handling subtraction of `d[i]` slightly suboptimally. Qualitatively, optimizing this code by hand can be difficult due to a lack of readability and the difficulty of debugging low-level code. The Autocomp-generated code is well-commented and the sequence of optimizations applied can be easily understood. This will be helpful for further optimization of this benchmark or future optimization of other benchmarks, via methods like the schedule reuse demonstrated in **??**.

*F. TinyMPC Primal Update Forward Pass Unoptimized Software ISA-Based Code Example*

```c
void test(float Adyn[12][12], float Bdyn[12][4] float Kinf[4][12], float x[NHORIZON + 1][12][1], float d[NHORIZON][4][1], float u[NHORIZON][4][1]) {
    static elem_t Kinf_x[4][1];
    static elem_t A_x[12][1];
    static elem_t B_u[12][1];

    for (int i = 0; i < NHORIZON; i++) {
        // define spad addresses for cached matrices
        // spad is row addressed and each row is 4 elements wide
        static uint32_t A_sp_addr = 0; // 144 elements, 0 to 35
        static uint32_t B_sp_addr = 36; // 48 elements, 36 to 47
        static uint32_t Kinf_sp_addr = 48; // 48 elements, 48 to 59
        static uint32_t C1_sp_addr = 60; // 16 elements, 60 to 63
        static uint32_t C2_sp_addr = 64; // 144 elements, 64 to 99
        static uint32_t x_sp_addr = 100; // 12 elements (at a time), 100 to 111
        static uint32_t u_sp_addr = 112; // 12 elements (at a time), 112 to 123
        static uint32_t acc_start_addr = 1 << 31;

        // tiled_matmul_spad_dram(Kinf, x[i], Kinf_x, NINPUTS, false, false);
        config_ex(WEIGHT_STATIONARY, NO_ACTIVATION, 1, false, false);
        config_st(4, 1.0);
        config_ld(48, 1.000000, 4, 0);
        config_ld(4, 1.000000, 4, 1);
        config_ld(4, 1.000000, 4, 2);
        mvin(Kinf, Kinf_sp_addr, 12, 4);
        mvin2(x[i][0], x_sp_addr, 1, 4);
        preload(x_sp_addr, acc_start_addr, 1, 4, 1, 4);
        compute_preloaded(Kinf_sp_addr, 0xffffffff, 4, 4, 4, 4);
        mvin2(x[i][4], x_sp_addr + 4, 1, 4);
        preload(x_sp_addr + 4, acc_start_addr | (1 << 30), 1, 4, 1, 4);
        compute_preloaded(Kinf_sp_addr + 4, 0xffffffff, 4, 4, 4, 4);
        mvin2(x[i][8], x_sp_addr + 8, 1, 4);
        preload(x_sp_addr + 8, acc_start_addr | (1 << 30), 1, 4, 1, 4);
        compute_preloaded(Kinf_sp_addr + 8, 0xffffffff, 4, 4, 4, 4);
        mvout(Kinf_x[0], acc_start_addr | (1 << 30), 1, 4);
        fence();

        static acc_t Kinf_x_negated[4][1] row_align_acc(1);
        static acc_t d_i_negated[4][1] row_align_acc(1);
        negate_matrix(Kinf_x, Kinf_x_negated, 4, 1);
        negate_matrix(d[i], d_i_negated, 4, 1);
        add_matrix(Kinf_x_negated, d_i_negated, u[i], 4, 1);

        // tiled_matmul_spad_dram(Adyn, x[i], A_x, NSTATES, false, false);
        config_ex(WEIGHT_STATIONARY, NO_ACTIVATION, 1, false, false);
        config_st(4, 1.0);
        config_ld(48, 1.000000, 4, 0);
        config_ld(4, 1.000000, 4, 1);
        config_ld(4, 1.000000, 4, 2);
        for (int chunk = 0; chunk < 3; chunk++)
            mvin(Adyn[chunk*4], A_sp_addr + chunk*12, 12, 4);
        mvin2(x[i][0], x_sp_addr, 1, 4);
        mvin2(x[i][4], x_sp_addr + 4, 1, 4);
        mvin2(x[i][8], x_sp_addr + 8, 1, 4);

        preload(x_sp_addr, acc_start_addr, 1, 4, 1, 4);
        compute_preloaded(A_sp_addr, 0xffffffff, 4, 4, 4, 4);
        preload(0xffffffff, acc_start_addr + 4, 1, 4, 1, 4);
        compute_accumulated(A_sp_addr + 12, 0xffffffff, 4, 4, 4, 4);
        preload(0xffffffff, acc_start_addr + 8, 1, 4, 1, 4);
        compute_accumulated(A_sp_addr + 24, 0xffffffff, 4, 4, 4, 4);

        preload(x_sp_addr + 4, acc_start_addr | (1 << 30), 1, 4, 1, 4);
        compute_preloaded(A_sp_addr + 4, 0xffffffff, 4, 4, 4, 4);
        preload(0xffffffff, (acc_start_addr + 4) | (1 << 30), 1, 4, 1, 4);
        compute_accumulated(A_sp_addr + 4 + 12, 0xffffffff, 4, 4, 4, 4);
        preload(0xffffffff, (acc_start_addr + 8) | (1 << 30), 1, 4, 1, 4);
        compute_accumulated(A_sp_addr + 4 + 24, 0xffffffff, 4, 4, 4, 4);

        preload(x_sp_addr + 8, acc_start_addr | (1 << 30), 1, 4, 1, 4);
        compute_preloaded(A_sp_addr + 8, 0xffffffff, 4, 4, 4, 4);
        preload(0xffffffff, (acc_start_addr + 4) | (1 << 30), 1, 4, 1, 4);
        compute_accumulated(A_sp_addr + 8 + 12, 0xffffffff, 4, 4, 4, 4);
        preload(0xffffffff, (acc_start_addr + 8) | (1 << 30), 1, 4, 1, 4);
        compute_accumulated(A_sp_addr + 8 + 24, 0xffffffff, 4, 4, 4, 4);

        mvout(A_x[0], acc_start_addr, 1, 4);
        mvout(A_x[4], acc_start_addr + 4, 1, 4);
        mvout(A_x[8], acc_start_addr + 8, 1, 4);
        fence();

        // tiled_matmul_spad_dram(Bdyn, u[i], B_u, NSTATES, false, false);
        config_ex(WEIGHT_STATIONARY, NO_ACTIVATION, 1, false, false);
        config_st(4, 1.0);
        config_ld(16, 1.000000, 4, 0);
        config_ld(4, 1.000000, 4, 1);
        config_ld(4, 1.000000, 4, 2);
        for (int chunk = 0; chunk < 3; chunk++)
            mvin(Bdyn[chunk*4], B_sp_addr + chunk*4, 4, 4);
        mvin2(u[i][0], x_sp_addr, 1, 4);
        preload(x_sp_addr, acc_start_addr, 1, 4, 1, 4);
        compute_preloaded(B_sp_addr, 0xffffffff, 4, 4, 4, 4);
        preload(0xffffffff, acc_start_addr + 4, 1, 4, 1, 4);
        compute_accumulated(B_sp_addr + 4, 0xffffffff, 4, 4, 4, 4);
        preload(0xffffffff, acc_start_addr + 8, 1, 4, 1, 4);
        compute_accumulated(B_sp_addr + 8, 0xffffffff, 4, 4, 4, 4);
        mvout(B_u[0], acc_start_addr, 1, 4);
        mvout(B_u[4], acc_start_addr + 4, 1, 4);
        mvout(B_u[8], acc_start_addr + 8, 1, 4);
        fence();

        add_matrix(A_x, B_u, x[i+1], 12, 1);}}
```

Fig. 22: Unoptimized software ISA-based starting code for the TinyMPC primal update forward pass from Sec. IV-E. Achieves 5.7% of theoretical maximum utilization.

## G. TinyMPC Primal Update Forward Pass Optimized Hardware FSM-Based Code Example

```
void test(float Adyn[12][12], float Bdyn[12][4] float Kinf[4][12], float x[NHORIZON + 1][12][1], float d[NHORIZON][4][1], float u[NHORIZON][4][1]) {
    static elem_t B_u[12][1];

    gemmini_extended_config_ex(1, 0, 0, 1, false, false);
    gemmini_extended3_config_ld(4, 1.0, false, 1);
    gemmini_extended3_config_ld(4, 1.0, false, 2);
    for (int i = 0; i < NHORIZON; i++)
    {
        gemmini_extended_config_st(4, 0, -1.0);
        gemmini_extended3_config_ld(48, 1.0, false, 0);
        gemmini_loop_ws(1, 1, 3, 0, 3, 0, Kinf, x[i], d[i], u[i], 12, 1, 1, 1, false, false, false, false, true, 0, 1, 1, false);
        gemmini_fence();

        if (i < NHORIZON - 1) {
            gemmini_extended_config_st(4, 0, 1.0);
            gemmini_extended3_config_ld(16, 1.0, false, 0);
            gemmini_loop_ws(3, 1, 1, 0, 3, 0, Bdyn, u[i], NULL, B_u, 4, 1, 1, 1, false, false, false, false, false, 0, 1, 1, false);
            gemmini_fence();

            gemmini_extended3_config_ld(48, 1.0, false, 0);
            gemmini_loop_ws(3, 1, 3, 0, 3, 0, Adyn, x[i], B_u, x[i+1], 12, 1, 1, 1, false, false, false, false, true, 0, 1, 1, false);
            gemmini_fence();
        }
    }
}
```

Fig. 23: Example of the TinyMPC primal update forward pass from Fig. 22, hand-optimized and using the hardware FSM, from the experiments. Achieves 10.9% of theoretical maximum utilization.

## H. TinyMPC Primal Update Forward Pass Autocomp-Generated Code Example

```c
void test(float Adyn[12][12], float Bdyn[12][4], float Kinf[4][12], float x[NHORIZON + 1][12][1], float d[NHORIZON][4][1], float u[NHORIZON][4][1]) {

  // Define scratchpad addresses for all matrices.
  uint32_t A_sp_addr      = 0;
  uint32_t B_sp_addr      = 36;
  uint32_t Kinf_sp_addr   = 48;
  uint32_t x_sp_addr      = 100;
  uint32_t acc_start_addr = 1 << 31;  // MSB = 1 indicates accumulator address

  //----------------------------------------------------------------
  // Hoisted Invariant Matrix Loads: These matrices do not change over
  // the horizon. We load them once into the Gemmini scratchpad.
  //----------------------------------------------------------------
  config_ex(WEIGHT_STATIONARY, NO_ACTIVATION, 1, false, false);
  config_st(4, 1.0);

  // Load constant weight matrix Kinf.
  config_ld(48, 1.000000, 4, 0);
  mvin(Kinf, Kinf_sp_addr, 12, 4);

  // Load constant system dynamics matrix Adyn.
  config_ld(48, 1.000000, 4, 0);
  for (int chunk = 0; chunk < 3; chunk++) {
    mvin(Adyn[chunk * 4], A_sp_addr + chunk * 12, 12, 4);
  }

  // Load constant control dynamics matrix Bdyn.
  config_ld(16, 1.000000, 4, 0);
  for (int chunk = 0; chunk < 3; chunk++) {
    mvin(Bdyn[chunk * 4], B_sp_addr + chunk * 4, 4, 4);
  }

  //----------------------------------------------------------------
  // Outer horizon loop with fusion of iterations and operations.
  //----------------------------------------------------------------
  for (int i = 0; i < NHORIZON; i++) {
    // Compute u[i] = -(Kinf*x[i]) - d[i]
    config_ld(4, -1.000000, 4, 1);
    mvin2(x[i][0], x_sp_addr, 1, 4);
    preload(x_sp_addr, acc_start_addr, 1, 4, 1, 4);
    compute_preloaded(Kinf_sp_addr, 0xffffffff, 4, 4, 4, 4);

    mvin2(x[i][4], x_sp_addr + 4, 1, 4);
    preload(x_sp_addr + 4, acc_start_addr | (1 << 30), 1, 4, 1, 4);
    compute_preloaded(Kinf_sp_addr + 4, 0xffffffff, 4, 4, 4, 4);

    mvin2(x[i][8], x_sp_addr + 8, 1, 4);
    preload(x_sp_addr + 8, acc_start_addr | (1 << 30), 1, 4, 1, 4);
    compute_preloaded(Kinf_sp_addr + 8, 0xffffffff, 4, 4, 4, 4);

    mvin2(d[i], x_sp_addr, 1, 4);
    config_ld(4, 1.000000, 4, 1);
    mvin2(0, x_sp_addr + 4, 1, 4);
    preload(x_sp_addr + 4, acc_start_addr | (1 << 30), 1, 4, 1, 4);
    compute_accumulated(x_sp_addr + 4, x_sp_addr, 1, 4, 1, 4);

    mvout(u[i], acc_start_addr, 1, 4);

    if (i < NHORIZON - 1) {
      // Compute A_x = Adyn * x[i]
      mvin2(x[i][0], x_sp_addr, 1, 4);
      mvin2(x[i][4], x_sp_addr + 4, 1, 4);
      mvin2(x[i][8], x_sp_addr + 8, 1, 4);

      preload(x_sp_addr, acc_start_addr, 1, 4, 1, 4);
      compute_preloaded(A_sp_addr, 0xffffffff, 4, 4, 4, 4);
      preload(0xffffffff, acc_start_addr + 4, 1, 4, 1, 4);
      compute_accumulated(A_sp_addr + 12, 0xffffffff, 4, 4, 4, 4);
      preload(0xffffffff, acc_start_addr + 8, 1, 4, 1, 4);
      compute_accumulated(A_sp_addr + 24, 0xffffffff, 4, 4, 4, 4);

      preload(x_sp_addr + 4, acc_start_addr | (1 << 30), 1, 4, 1, 4);
      compute_preloaded(A_sp_addr + 4, 0xffffffff, 4, 4, 4, 4);
      preload(0xffffffff, (acc_start_addr + 4) | (1 << 30), 1, 4, 1, 4);
      compute_accumulated(A_sp_addr + 4 + 12, 0xffffffff, 4, 4, 4, 4);
      preload(0xffffffff, (acc_start_addr + 8) | (1 << 30), 1, 4, 1, 4);
      compute_accumulated(A_sp_addr + 4 + 24, 0xffffffff, 4, 4, 4, 4);

      preload(x_sp_addr + 8, acc_start_addr | (1 << 30), 1, 4, 1, 4);
      compute_preloaded(A_sp_addr + 8, 0xffffffff, 4, 4, 4, 4);
      preload(0xffffffff, (acc_start_addr + 4) | (1 << 30), 1, 4, 1, 4);
      compute_accumulated(A_sp_addr + 8 + 12, 0xffffffff, 4, 4, 4, 4);
      preload(0xffffffff, (acc_start_addr + 8) | (1 << 30), 1, 4, 1, 4);
      compute_accumulated(A_sp_addr + 8 + 24, 0xffffffff, 4, 4, 4, 4);

      // Compute B_u = Bdyn * u[i] and accumulate onto A_x
      mvin2(u[i][0], x_sp_addr, 1, 4);
      preload(x_sp_addr, acc_start_addr | (1 << 30), 1, 4, 1, 4);
      compute_preloaded(B_sp_addr, 0xffffffff, 4, 4, 4, 4);
      preload(0xffffffff, (acc_start_addr + 4) | (1 << 30), 1, 4, 1, 4);
      compute_accumulated(B_sp_addr + 4, 0xffffffff, 4, 4, 4, 4);
      preload(0xffffffff, (acc_start_addr + 8) | (1 << 30), 1, 4, 1, 4);
      compute_accumulated(B_sp_addr + 8, 0xffffffff, 4, 4, 4, 4);

      mvout(x[i + 1][0], acc_start_addr, 1, 4);
      mvout(x[i + 1][4], acc_start_addr + 4, 1, 4);
      mvout(x[i + 1][8], acc_start_addr + 8, 1, 4);

      fence();
    }
  }
}
```

Fig. 24: Example of an Autocomp-optimized version of the TinyMPC primal update forward pass from Fig. 22. Achieves 15.7% of theoretical maximum utilization.

