# OpenReview forum: "Autocomp: LLM-Driven Code Optimization for Tensor Accelerators"
_iscaconf.org/ISCA/2025/Workshop/MLArchSys — MLArchSys 2025 Oral_

### Official Review · Reviewer_u8HC · 2025-05-12
**This paper introduces Autocomp that LLMs to automatically optimize accelerator code, specifically for tensor accelerators.**

**Confidence:** 3
**Rating:** 6

**Detailed Feedback And Questions For Authors:**

This paper presents a new approach for automatically optimizing low-level tensor accelerator code using LLMs. The proposed system, Autocomp, introduces a two-phase prompting mechanism and integrates performance feedback into a beam search-based exploration loop. The results are showing consistent improvements over compiler-generated, expert-tuned, and even hardware FSM implementations across matrix multiplication, convolution, and robotics workloads. The methodology is reasonably engineered and the evaluation is hardware-grounded, which makes the contribution practical. That said, a few aspects of the paper could be clarified or strengthened to better support its claims.

My first concern is about the motivation for using LLMs specifically for this task. While the paper demonstrates that LLMs can be effective, it does not sufficiently explain why LLMs are a better (if we consider their downsides as well) choice than other alternatives.

My second concern is about the workload complexity and generality. I understand that the current workloads are great and indeed needed for the proof of concept and the very first steps of designing the system to see if the opportunity exists.However, My concern is that since matrix multiplication and 2D convolution are widely studied and often have well-known optimization patterns, can using them proof how and how much an LLM-based is capable of optimizing the difficult cases. While the robotics kernel adds some variety (which I really liked), the overall workload suite lacks irregular or less structured tasks where optimization strategies are less obvious.

My last and again less major comment is about interpretability and insight into optimization behaviors. The system performs well, but it remains a bit of a black box. It would strengthen the contribution to include some analysis or categorization of the kinds of optimizations Autocomp repeatedly discovers, or where it diverges from traditional heuristics.

**Top Reasons To Accept The Paper:**

This is an interesting and timely paper.

1- First of all, the prelim results are promising as they show that Autocomp consistently outperforms both compiler-generated and expert hand-tuned code, and surpasses a hardware FSM in a robotics use case.

2- Evaluation-wise, the two-stage prompting, feedback-guided optimization, and use of FireSim for cycle-accurate evaluation show thoughtful system design and realism. Comparisons include unoptimized software, compiler-generated code (Exo), hand-tuned expert code, and FSM hardware, providing context for Autocomp’s effectiveness.

3- I think the success metrics are sufficient. The use of latency, utilization, and correctness as evaluation criteria is justified and reported.

**Top Reasons To Reject The Paper:**

The following aspect of the paper can be improved:

1- Probably this is a weakness but not directly a reason for rejection: While LLMs are used effectively, the paper lacks a rigorous motivation for why LLMs are preferable over other learned or heuristic optimization approaches. This is a timely topic, of course, but regardless of the popularity of LLMs, I think an individual paper should still include a stand-alone concrete motivation.

2- My more major issue is about the workloads, their complexity, and the level of difficulty to have optimized code for them. Matrix multiplication and 2D convolution are well-studied and probably more straightforward to optimize; the paper could benefit from more varied or less regular workloads to further demonstrate generality and cases that LLMs are going to be really needed.

3- Finally, the paper doesn’t deeply analyze why certain LLM-generated optimizations work better. This, IMO, is needed to share some insight into what Autocomp learns or discovers.

---

### Official Review · Reviewer_wNWR · 2025-05-15
**Review of Autocomp**

**Confidence:** 4
**Rating:** 6

**Detailed Feedback And Questions For Authors:**

[author anonymity concern] The first contribution in the paper says "We devise a two-stage prompting technique that enables us to 1) generate salient plans to optimize DSL code for our tensor accelerator, Gemmini, and 2) implement those plans as rewrites to the original code." -> "our tensor accelerator, Gemmini" suggests that this is the authors' prior work. Please watch out for maintaining author anonymity and review double-blindedness (or if this is different work, please change the wording of this sentence so it is not misleading).

This paper presents Autocomp, an LLM-assisted approach to figure out optimization passes to apply to accelerator DSL code. This is done through careful prompting for the LLM in two stages. The authors demonstrate their proof of concept by showing that Autocomp generates 1.34X faster matrix multiplication and 1.03X 2D convolution code than expert hand-tuned code and 1.51X faster code for robotics kernels that outperforms a hardware implementation.

I think this paper is suitable for a workshop venue. Here are some questions I have that I think should be included in the workshop paper:

- What is the overhead of your approach? The authors mention that there are two phases, which sound fairly involved -- they have to check correctness with a functional test suite, which likely takes some time, and then they need to run an RTL simulation to get the latency, which also likely takes time. Please quantify the overhead of the approach in the workshop paper for the ResNet-50 example you present in the paper. It would also be good to know how this overhead would scale for transformer models, which are typically larger than ResNet-50.

- The paper mentions that only functionally equivalent candidates are select and incorrect code generation options are dropped. What is the typical rate of incorrect (non-functional) code generation by the LLM during the optimization process?  How robust is the functional test suite in catching these errors? Please add this rate and any discussion of the test suite missing errors to the paper.

- The authors claim a 1.51X speedup for the robotics application over a "expert hand-tuned hardware FSM-based implementation" baseline, but there is not a reference to a paper or explanation of what this hand-tuning entailed, so it is difficult to understand what this baseline did and the context surrounding this comparison. Please add details about this baseline so the reader can properly evaluate your contribution. In a future extension of the paper, it would be helpful to compare to previously-published baselines so you can reference those papers for the reader.

- How extensible is this approach to accelerators and systems other than Gemmini? What would be the main challenges in using Autocomp with a different hardware accelerator? In particular, how would the prompts themselves need to change?

- The paper describes how the prompts are structured. How sensitive is the performance of Autocomp to the precise wording and structure of the prompts? How much of a tuning process was there in designing effective prompts that got to the results presented in the paper?

Smaller questions / notes:

- Transformers are particularly emphasized in the introduction, but only ResNet-50 is evaluated at the end. I think this is okay for a workshop venue, but please tone down the emphasis of transformers and attention in the introduction.

- A 1.03X improvement for 2D convolutions is small and I do not think it needs to be emphasized in the abstract, introduction, and conclusion. I would suggest focusing on the other two numbers (1.34X and 1.51X) instead.

- Some references that I thought may be relevant to your accelerator code optimization section (please read and decide whether you think they are relevant):
[1] AutoDNNchip: An automated DNN chip predictor and builder for both FPGAs and ASICs. FPGA' 20. -> they look at producing synthesizable RTL code with optimized algorithm-to-hardware mapping
[2] AHA: An Agile Approach to the Design of Coarse-Grained Reconfigurable Accelerators and Compilers, TECS'22 -> they consider code optimizations in a Halide to accelerator flow

- Citations 6 and 10 should have the updated name of the conference, NeurIPS.

Other questions I have that I think should be included in a full paper extension of this workshop paper:

- How does Autocomp compare to the optimization capabilities of more traditional compilers like TVM or XLA, particularly in terms of the types of optimizations they can discover and apply? The paper mentions these compilers but a direct comparison of optimization capabilities would be helpful, especially if the optimization suite is expanded in a full paper.

- Please expand your evaluation to larger models and transformers past ResNet-50. In particular, discussing how the overhead of your approach scales when targeting larger models and transformers would be interesting.

- It would be helpful to compare to previously-published hardware implementation baselines so you can reference those papers for the reader and the reader can then evaluate your contributions within that context.

**Top Reasons To Accept The Paper:**

- Working on compilers for ML accelerators is a timely area. The authors proposal of integrating LLMs to automate this workflow is a valuable research direction.
- The authors' proof of concept, Autocomp, generates code that is 1.34X faster for matrix multiplication, 1.03X for 2D convolutions, and 1.51X faster for robotics kernels, with the robotics application also resulting in a higher hardware utilization compared to the baseline.
- Paper is clearly written and easy to follow.

**Top Reasons To Reject The Paper:**

- Author anonymity concern (explained below in the detailed feedback section)
- No mention or discussion on the overhead of this approach. This approach requires checking correctness with a functional test suite and then an RTL simulation for latency estimates, which can both be computationally expensive, especially since this needs to be tested for many different approaches, but the authors do not mention what the runtime or cost of this approach is in the evaluation.
- Baseline for the 1.51X speedup over a hardware implementation does not have details or a paper reference, so it is difficult to evaluate this contribution without the context of what the baseline implementation did.

---

### Official Review · Reviewer_sKhi · 2025-05-18
**Detailed and Interesting Paper**

**Confidence:** 4
**Rating:** 7

**Detailed Feedback And Questions For Authors:**

This is an interesting and well-written paper. In particular, I am happy that the authors shared detailed information about their prompt and implementation. A few small questions remain:

1. You mention the monetary cost of your optimizations in the conclusion. Are you able to share data regarding these costs?
2. In III.E, you mention ensembling the o3-mini and gpt-4o models. Are you able to provide a qualitative or quantitative basis for how the models' outputs differ? Is one consistently better or worse?
3. Assuming the comments in Fig. 11 represent the optimizer's reasoning, I would be interested to see a column to the right or left with separate comments explaining the optimizations. Which are good, which are bad? Is there anything in Fig. 11 that is uniquely exciting? What rules and steps did Autocomp take to get to the code in Fig. 11? How much did each of these steps speed up the code?
4. II.B mentions that the LLM takes you beyond the nested loop optimization/transform abstraction. Are you able to (at least qualitatively) discuss some of the optimization steps that were performed in Fig. 11 that _could not_ have easily been performed by a set of nested-loop rewrite rules working on a higher-level IR? Even if Autocomp is only doing nested-loop transformation with LLMs, the work is still interesting, but it would be far more interesting if you could demonstrate something more complicated.

**Top Reasons To Accept The Paper:**

Automatic optimization of code, especially legacy code and code written for esoteric architectures, is a very relevant topic. The paper provides a significant amount of detail on the Gemmini optimization process, including detailed prompts and samples of what the LLM is capable of.

**Top Reasons To Reject The Paper:**

None